# Repurposing statins and phenothiazines to treat chemoresistant neuroblastoma

Katarzyna Radke [iD][1], Kristina Aaltonen[1], Erick A Muciño-Olmos[1], Javanshir Esfandyari[1],
Aleksandra Adamska[1], Joachim T Siaw [iD][1], Dora Adamic [iD][1], Chiara Lago [iD][1], Adriana Mañas[1,2,3],
Alexandra Seger[1], Karin Hansson [iD][1,4], Oksana Rogova [iD][5], Sophie Lehn [iD][1], Daniel J Mason[6],
Daniel J O'Donovan [iD][6], Ian Roberts [iD][6], Antonia Lock [iD][6,10], Jane Brennan [iD][6], Kristian Pietras [iD][1],
Emma J Davies [iD][6], Peter Spégel[5], Oscar C Bedoya-Reina[7,8], David Brown[6], Neil T Thompson[6,11],
Cesare Spadoni[9] & Daniel Bexell [iD][1✉]

## Abstract

**Relapse and treatment resistance are common in children with high-risk neuroblastoma, and novel therapies are needed. Conventional drug discovery is slow, expensive, often fails in practice, and consequently falls short in addressing pediatric and rare conditions. In such instances, drug repurposing is a promising strategy. Here, we used two independent in silico prediction tools including machine learning to identify approved drugs for repurposing against neuroblastoma. The combination of statins and phenothiazines showed strong synergistic effects in human neuroblastoma organoids, decreased tumor growth, and prolonged survival in *MYCN*-amplified neuroblastoma patient-derived xenografts. The drug combination altered cholesterol metabolism through two different mechanisms and induced a phenotypic change toward an adrenergic state in vitro, which was associated with enhanced chemosensitivity. Integration of the drug combination into standard-of-care chemotherapy regressed tumors and prolonged survival in chemoresistant patient-derived xenografts. Thus, a combination of safe and approved medications added to standard-of-care chemotherapy outperforms chemotherapy alone in chemoresistant neuroblastoma.**

**Keywords** Drug Repurposing; Machine Learning; Neuroblastoma; Phenothiazine; Statin
**Subject Category** Cancer

## Introduction

The rarity of pediatric cancer and ethical considerations specific to children have hindered drug development for this group of patients (Nishiwaki and Ando, 2020; Pearson et al, 2022). One option to progress therapies for rare diseases is drug repurposing, where an approved drug with human safety and efficacy data is given for an alternative, unapproved indication, which is a time-saving and cost-effective approach. Patient groups with limited treatment options can especially benefit from this approach. Furthermore, efficacy and safety data on existing drugs can provide the basis for computational predictions and help with mechanistic rationalization in screening approaches and preclinical and clinical trials (Pantziarka et al, 2021; van den Berg et al, 2021).

Neuroblastoma (NB) is a childhood malignancy of the sympathetic nervous system responsible for nearly 15% of pediatric cancer-related deaths. NB is a clinically heterogeneous disease, spontaneously regressing with minimal intervention in some children but pursuing an aggressive metastatic course in others. Patients with high-risk NB undergo intensive chemotherapy, surgery, radiotherapy, and other treatments, and, although initial treatment can be effective, over 50% of patients with high-risk NB eventually relapse with treatment-resistant disease. Thus, there is an unmet need for new and better treatment approaches for children with high-risk NB (Park et al, 2017; Maris, 2010; Cohn et al, 2009).

The molecular pathology of NB is characterized by copy number aberrations (CNAs), including 1p loss, *MYCN* amplification, 11q loss, 17q gain, and others (Pugh et al, 2013). Relapsed tumors are enriched for mutations in genes activating oncogenic signaling pathways, such as the RAS-mitogen-activated protein kinase (MAPK) and the YAP-Hippo pathways (Eleveld et al, 2015;

---

[1]Translational Cancer Research, Lund University, Lund, Sweden. [2]Translational Research in Pediatric Oncology, Hematopoietic Transplantation and Cell Therapy, IdiPAZ Research Center, University Hospital La Paz, Madrid, Spain. [3]Pediatric Onco-hematology Clinical Unit IdiPAZ-CNIO, National Cancer Research Center (CNIO), Madrid, Spain. [4]Cancer Stem Cell Laboratory, The Breast Cancer Now Toby Robins Research Centre, The Institute of Cancer Research, London, UK. [5]Department of Chemistry, Centre for Analysis and Synthesis, Lund University, Lund, Sweden. [6]Healx Ltd., Charter House, 66-68 Hills Road, Cambridge CB2 1LA, UK. [7]School of Medical Sciences, Örebro University, Örebro, Sweden. [8]Childhood Cancer Research Unit, Department of Women's and Children's Health, Karolinska Institutet, Stockholm, Sweden. [9]aPODD Foundation, Virginia Cottage, Bridgnorth Road, Stourton, Stourbridge, West Midlands DY7 5BQ, UK. [10]Present address: EMBL-EBI, Wellcome Genome Campus, Hinxton, Cambridgeshire CB10 1SD, UK. [11]Deceased: Neil T Thompson. ✉E-mail: daniel.bexell@med.lu.se

Schramm et al, 2015). However, compared with many adult tumors, NB has a relatively low mutational burden and few recurrent mutations. While genetic changes, especially CNAs, are important for NB initiation and progression, it has become increasingly clear that transcriptional activity and phenotypic plasticity are important as the disease progresses and fails to respond to treatment (Ponzoni et al, 2022).

Epigenetic and transcriptional analyses have defined two NB cell states in vitro; a lineage-committed adrenergic (ADR or ADRN) state, and an immature neural crest cell-like or mesenchymal-like (MES) state (Van Groningen et al, 2017; Boeva et al, 2017). These cell states exist along a continuum with a high degree of plasticity. Current data suggest that both cell states are important for NB progression, with the MES-like state involved in chemoresistance and relapse (Van Groningen et al, 2017; Boeva et al, 2017; van Groningen et al, 2019; Bedoya-Reina et al, 2021; Gartlgruber et al, 2021; Yuan et al, 2022; Olsen et al, 2024; Mañas et al, 2022; Sengupta et al, 2022; Thirant et al, 2023; Patel et al, 2024).

Disease-gene expression matching (DGEM) represents a methodology aimed at identifying potential therapeutic drugs for diseases characterized by specific gene expression signatures (Tranfaglia et al, 2019). By comparing genes differentially expressed in disease states with the gene expression profiles induced by various drugs, existing pharmaceuticals suitable for repurposing can be identified. This approach holds particular relevance for conditions such as NB, where transcriptomic alterations play a critical role in disease progression (Iorio et al, 2013; Pushpakom et al, 2019).

Another prediction algorithm, PRISM (Predicted Repurposed Indications from Similarity Matrices), can predict novel treatment indications based on known treatment indications and similarity measures between compounds and diseases using machine learning. These measures are obtained by comparing the compounds' structures or examining their target similarities. In addition, PRISM defines the similarity between diseases by analyzing ontological data, disease targets, and the frequencies with which diseases co-occur in literature (Zhang et al, 2014). Here, PRISM was used to find new drug–disease associations for NB.

We harnessed the power of DGEM and PRISM prediction tools to identify potential candidates for drug repurposing in high-risk NB. This yielded several drug hits already approved for non-cancer diseases. The efficacies of candidate drugs were validated in human high-risk NB organoids, conventional cell lines, and in patient-derived xenograft (PDX) models. A combination of selected classes of drugs resulted in alterations in cholesterol metabolism, synergistic effects, and NB cell state shift toward a chemosensitive ADR-like state in vitro. Notably, treatment with the novel drug combination alongside standard-of-care (COJEC) chemotherapy resulted in tumor regression and extended survival compared with standard-of-care chemotherapy alone in a chemoresistant NB PDX model, paving the way for testing of this combination towards clinical practice.

# Results

We identified drugs for repurposing in high-risk NB using in silico prediction tools, including machine learning, validation of single drugs and drug combinations in human NB organoids,

transcriptional analysis of treatment-induced phenotype, lipidomics, and in vivo testing in NB chemoresistant PDX models (Fig. 1).

## In silico drug selection using DGEM and PRISM

Publicly available raw gene expression datasets were manually curated and analyzed for their suitability to reflect the transcriptomic profile of the disease. Four raw gene expression datasets passed quality control criteria that ensured robustness of the input data. These datasets included transcriptomic data from NB patients with associated clinicopathological information, including the disease stage, relapse status, and molecular determinants of risk such as *MYCN* amplification and 17q gain (Appendix Table S1). We applied the same rationale to define a query gene signature that encapsulates molecular descriptors of worsening disease. Within the context of NB biology, this is represented by patient clinical information of low risk vs. high risk, *MYCN* amplified vs. non-*MYCN*, and alive vs. dead. The intention was to survey multiple facets of NB biology within connectivity mapping, an approach that compares gene expression signatures to a reference database of drug- and perturbation-induced profiles to identify potential functional relationships between genes, compounds, and disease states. Gene expression profiles (GEPs) generated from differentially expressed genes in these datasets were evaluated by gene set enrichment analysis (GSEA) to ensure that they described biological processes indicative of an NB phenotype. All datasets had enrichments of genes belonging to terms in keeping with NB biology and clinical data of the studies used to define GEPs, such as MYC targets, E2F targets, glycolysis, oxidative phosphorylation, mTORC1 signaling, and proliferation (G2M checkpoint) (Appendix Fig. S1A). The NB GEPs obtained through this process were used for drug matching.

DGEM predicts drugs likely to be effective in a given disease based on the most differentially expressed genes in a GEP. Briefly, a disease GEP is compared to a compendium of thousands of public and private drug GEP to detect drug–disease associations. The dataset covers a range of pharmacological classes and includes a mixture of approved, experimental, and investigational compounds to guide repurposing recommendations. The strongest drug signatures may, for example, be inversely correlated with the disease signature, in which case the drug may elicit a therapeutic effect on the disease.

In an independent analysis, potential NB targeting drugs were also identified with a guilt-by-association approach using the PRISM machine-learning algorithm. Here, compound similarity measures like structural similarities and target homologies, as well as disease similarities retrieved from ontological data, target profiles, and literature co-occurrence measures were used for finding previously unexplored treatment candidates (Zhang et al, 2014; Wang et al, 2013).

Prediction results were first analyzed by automated computational tools at Healx, which provided annotations with supporting pharmacodynamic data from an internal knowledgebase relevant to inferring therapeutic benefit in NB, such as literature support, clinical trials, and drug mechanism of action. The initial automated ranking of over 1309 drugs approved by the FDA or EMA yielded 192 predictions comprising 150 unique compound names. At this point, the output was assessed by a human expert drug discoverer.

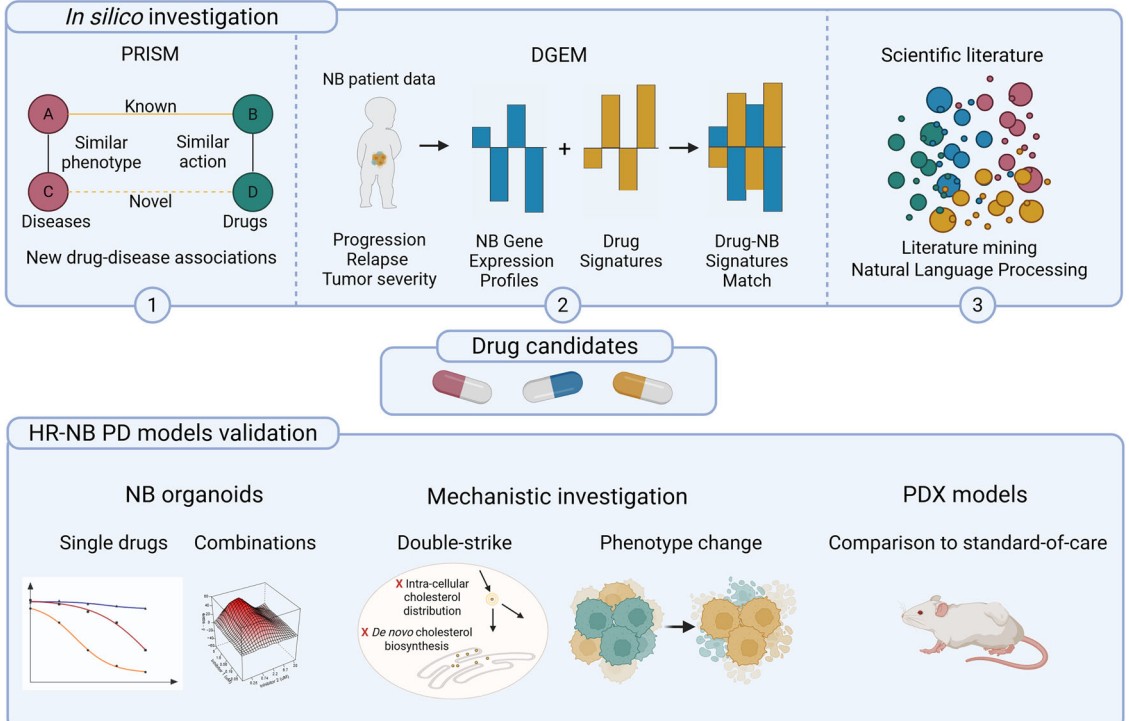

**Figure 1. Overview of the study.**

In silico, potential drugs for repurposing were identified using three complementary methods: (1) new drug–disease associations were found based on the PRISM (guilt-by-association) machine-learning algorithm, (2) drug-NB signature matches were identified with DGEM using neuroblastoma gene expression profiles of progression, tumor severity, or relapse, together with a repository of drug signatures, and (3) literature mining with natural language processing was used as a confirmation tool for hits commonly identified with the other methods. Drug candidates were subsequently tested in high-risk neuroblastoma (HR-NB) models as single drugs and in combination. Investigation of the mechanism of action of the selected drugs suggested two independent hits (double-strike) on cholesterol metabolism leading to cholesterol deficiency, as well as gene expression profile-based phenotype changes after combination treatment. The effect of the combination was investigated in vivo along with standard-of-care chemotherapy. HR-NB high-risk neuroblastoma, DGEM Disease-Gene Expression Matching, PRISM Prediction of Repurposed Indications with Similarity Matrices, PD models patient-derived models.

---

The 150 drugs were reviewed using several criteria which aim to identify drugs and mechanisms that could achieve the desired therapeutic effects for patients. Genes and biological pathways associated with NB were included. Safe drugs with favorable pharmacokinetic properties and minimal side-effects were prioritized. For example, drugs should be capable of absorption into the body, be slowly metabolized, and safely cleared. This triage process led to the selection of the 12 compounds listed in Table 1.

## Testing of single drugs in NB organoids

Selected agents were tested as single drugs in three human NB organoid models (LU-NB-1, LU-NB-2, and LU-NB-3) derived from *MYCN*-amplified high-risk PDX tumors (Appendix Table S2) (Persson et al, 2017). These PDX models retain the genotype, transcriptional profiles, and phenotypes of their corresponding patient tumors, and they represent a heterogeneous population of tumor cells (Braekeveldt et al, 2018; Karlsson et al, 2024; Mañas et al, 2022). Four of the 12 drugs decreased the viability of all three NB organoid models (Fig. 2A; Appendix Fig. S1B-D): the three phenothiazines (used as antipsychotics, dopamine receptor antagonists), trifluoperazine (TFP), thioridazine (TRD), and prochlorperazine (PCZ), and the HMG-CoA reductase inhibitor lovastatin

(LOV) (Fig. 2A). The half-maximal inhibitory concentration ($IC_{50}$) and area under the curve (AUC) were calculated for these four drugs at 3 and 7 days (Appendix Table S3). The effect of lovastatin was higher on day 7 compared with day 3 (Appendix Table S3).

## Drug combinations are synergistic in NB organoids

Combination therapy with multiple drugs can reduce the development of resistance and reduce the risk of drug toxicity as lower concentrations of individual drugs are required (Doroshow and Simon, 2017). We hypothesized that combining the two most effective single-agent drugs with different mechanisms of action, cholesterol-lowering statins (LOV) and one of the phenothiazines (TFP, TRD, or PCZ), could be a promising combination approach.

We therefore tested combinations of LOV with either TFP or PCZ in the LU-NB-1 and LU-NB-2 organoid models, due to their chemoresistant characteristics in vitro and in vivo (Mañas et al, 2022). Combination treatment in LU-NB-1 and LU-NB-2, respectively, resulted in additive or synergistic ZIP scores for LOV + PCZ (2.84/3.24 at 3 days and 14.3/11.67 at 7 days) and for LOV + TFP (6.36/1.29 at 3 days and 5.03/17.34 at 7 days) (Fig. EV1A).

To confirm that synergistic effects were not specific to these drugs but more general to the two drug classes, we evaluated the

**Table 1. Predicted drugs tested in neuroblastoma organoids.**

| Drug | Drugbank ID | MW (g/mol) | Indication (examples) |
|---|---|---|---|
| Trifluoperazine (TFP) | DB00831 | 480.42 | Antipsychotic drug used in schizophrenia<br>Anti-adrenergic, antidopaminergic<br>Phenothiazine |
| Thioridazine (TRD) | DB00679 | 407.04 | Antipsychotic drug used in schizophrenia<br>Lower potency than trifluoperazine and withdrawn<br>Phenothiazine |
| Nortriptyline | DB00540 | 299.84 | Tricyclic antidepressant, norepinephrine reuptake inhibitor |
| Prochlorperazine (PCZ) | DB00433 | 606.09 | Antipsychotic drug,<br>D2 receptor antagonist<br>Antiemetic properties, Phenothiazine |
| Resveratrol | DB02709 | 228.25 | Stilbenoid, natural phenol<br>Dietary supplement |
| Sildenafil | DB00203 | 666.7 | Vasodilation used in e.g., Viagra |
| Dasatinib | DB01254 | 488.01 | Tyrosine kinase inhibitor, e.g. in CML |
| Pazopanib | DB06589 | 437.52 | RTK inhibitor, angiogenesis inhibition<br>Used in e.g., renal cell carcinoma |
| Lovastatin (LOV) | DB00227 | 404.54 | Cholesterol-lowering statin<br>Inhibitor of HMG-CoA reductase |
| Folic Acid | DB00158 | 441.4 | Vitamin B |
| Topiramate | DB00273 | 339.36 | Anti-convulsant |
| Tianeptine | DB09289 | 436.95 | Tricyclic antidepressant |

most synergistic area (MSA) and efficacy scores between phenothiazines (TFP and PCZ) and different statins (lovastatin (LOV), fluvastatin (FLV), and pitavastatin (PIT)). To select optimal drugs for further testing and translational relevance, we also examined the in vitro combinatory effects and pharmacokinetic features of the drugs together with their intended use. PCZ and PIT had one of the highest synergy scores and efficacy effects in LU-NB-1 and LU-NB-2 models both at day 3 and day 7 (Figs. 2B,C and EV1B). In addition, PIT had the highest bioavailability and favorable pharmacokinetics/pharmacodynamics of the statins (Juarez and Fruman, 2021).

The efficacy of PCZ and PIT was further assessed in non-*MYCN* amplified NB, by treating SK-N-AS and SK-N-SH cell lines with the selected drugs. Both single-agent (Fig. EV1C) and combination treatments (Fig. EV1D) reduced NB cell viability, indicating that the therapeutic response is independent of *MYCN* amplification.

Further, drug combination-induced cell death was also detected in NB organoids by visual inspection and an increase in PI-positive staining (dead cells) (Fig. 2D,E). Overall, treatment of chemoresistant NB in vitro models with statins in combination with phenothiazines resulted in strong drug synergy and NB cell death.

## Combined prochlorperazine and pitavastatin is effective in an NB PDX model

We next tested PCZ and PIT alone and in combination in vivo (Appendix Fig. S2A). Mice were randomly allocated to treatment groups when tumors reached at least 150 mm³. Mice were treated intraperitoneally (i.p.) with PCZ, PIT, and their combination (5 mg/kg each or drug vehicle daily, six times a week for 33 days) (Appendix Fig. S2A). The average tumor size at the start of treatment was ~200 mm³ (Appendix Fig. S2B). Treatment was well tolerated, with only transient drowsiness seen with PCZ and no

evidence of weight loss (Appendix Fig. S2C). However, i.p. administration of the drugs did not reduce tumor growth (Appendix Fig. S2D), and transcriptomic analysis of excised tumors did not reveal any group-specific clustering (Appendix Fig. S2E). Given these results, it was likely that i.p. delivery could not achieve effective drug concentrations in the tumors, possibly due to first-pass metabolism (Björkhem-Bergman et al, 2011).

We therefore moved forward using direct administration of the drugs (alone or in combination) into the tumor by intratumoral (i.t.) injection in the treatment-resistant and aggressive NB model LU-NB-1 (Fig. 2F). The LU-NB-1 model is resistant to standard NB chemotherapy treatment in vitro and in vivo (Mañas et al, 2022), reflecting the clinical scenario of a lack of response. LU-NB-1 PDX cells were injected subcutaneously into NSG mice, which were randomly allocated to control, single-drug treatment, or combination treatment groups when the tumors reached over 350 mm³ (Fig. EV1E). Mice were treated i.t. daily five times a week according to the group assignment (Fig. 2F) and were monitored for weight loss (Fig. EV1F), tumor growth, and survival (Figs. 2G and EV1G,H). Compared with control, treatment with the drug combination significantly delayed tumor growth (one-way ANOVA with Tukey's multiple comparisons test, $P = 0.049$) at day 8 of treatment and prolonged survival of these mice ($P = 0.001$, log-rank test between controls and combination) (Fig. 2G,H). Thus, while systemic administration of the PCZ + PIT drug combination did not affect tumor growth, intratumoral treatment decreased tumor growth and prolonged survival in a chemoresistant NB PDX model.

## Combined prochlorperazine and pitavastatin modulates cholesterol synthesis pathways

We performed transcriptomic analysis by RNA-seq to investigate the molecular mechanisms of action of the drug combination

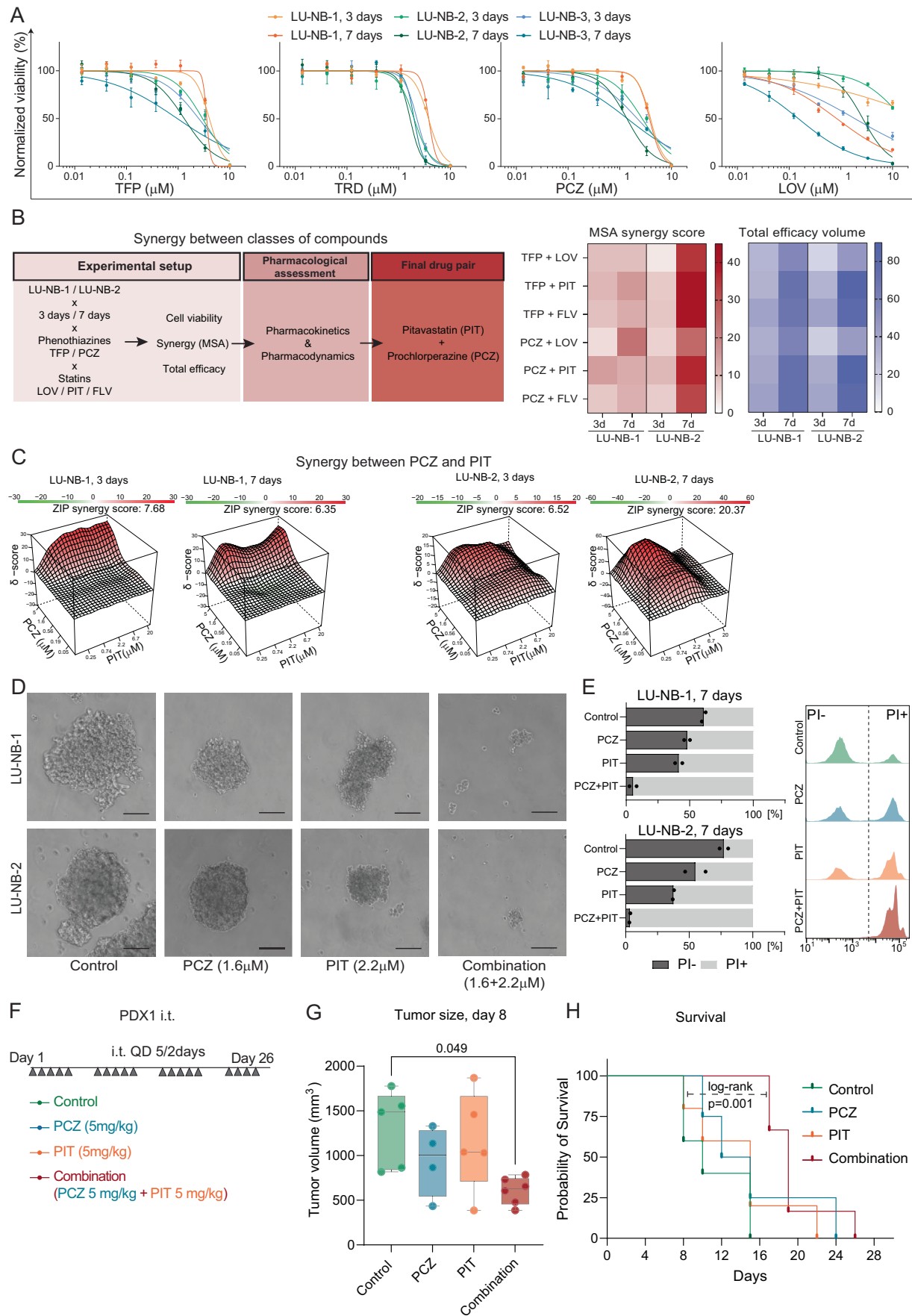

**Figure 2. Single drug efficacy and PCZ + PIT combination synergy.**

(A) Dose–response curves of the four top hits tested in three PDX-derived high-risk neuroblastoma organoids (LU-NB-1, 2, and 3) for 3 and 7 days, mean percent viability of control ±SD, $n = 3$. (B) Synergy testing between the two drug classes, phenothiazines and statins. Workflow for identifying the final drug pair. Combination total scores of 6 ×6 synergy matrices over 3 or 7 days, SynergyFinder's MSA score and total efficacy volume (total effect calculated under the whole dose–response matrix), $n = 2$. (C) Synergy landscape of the PCZ + PIT combination tested in LU-NB-1 and LU-NB-2 organoids for 3 and 7 days, average of $n = 2$. (D) Representative photographs of NB organoid size and integrity following single drug and PCZ (1.6 μM) + PIT (2.2 μM) combination treatment (scale bar 100 μm), 7 days. (E) Propidium iodide (PI) staining for PCZ (1.6 μM) + PIT (2.2 μM) combination over a 7-day treatment. Bars represent the mean, $n = 2$ per group. (F) Experimental set-up for intratumoral (i.t.) treatment with single drugs and the combination. Subcutaneous LU-NB-1 PDX tumors were established in NSG mice. (G) Average tumor volume per group, day 8 (control $n = 5$, PCZ $n = 4$, PIT $n = 5$, combination $n = 7$, one-way ANOVA followed by Tukey's multiple comparisons test; boxes represent the interquartile range and whiskers indicate minimum and maximum values). (H) Kaplan–Meier curves showing survival in days from the first treatment day. Statistical analysis was performed using the log-rank test between control and combination group. TFP trifluoperazine, TRD thioridazine, PCZ prochlorperazine, LOV lovastatin, PIT pitavastatin, FLV fluvastatin, MSA Most Synergistic Area score, ZIP zero interaction potency score, QD quaque die-once a day. Source data are available online for this figure.

in vitro. After 48 h, PIT and the combination treatment induced the greatest differential gene expression compared with controls, with PCZ producing more subtle changes (Appendix Fig. S3A,B). As expected from the previously described drug mechanisms, GSEA suggested that PCZ inhibits dopaminergic synapse responses, whereas PIT modulates cholesterol metabolism (Fig. 3A; Appendix Fig. S3C). Unexpectedly, PCZ treatment also increased the expression of genes involved in cholesterol and steroid metabolism. Both drugs were associated with downregulation of genes involved in DNA replication (Fig. 3A; Appendix Fig. S3C). Analysis of KEGG pathways indicated that several biosynthesis processes, including the steroid and terpenoid backbone and lipid metabolism, were strongly upregulated (Fig. 3A; Appendix Fig. S3C). Downregulation of genes involved in drug and nucleotide metabolism and oxidative phosphorylation was consistent with a reduction in normal proliferation (Appendix Fig. S3C).

Transcription factor (TF) activity was inferred using CollecTRI, which leverages high-confidence TF–target interactions to estimate regulatory activity from gene expression data. This analysis revealed that combination drug treatment led to markedly reduced activity of MYC and E2F, consistent with decreased NB cell proliferation (Fig. 3B). These findings were further supported by downregulation of phosphorylated Akt protein following combination treatment (Appendix Fig. S3D). Interestingly, both single and combined treatments induced activation of downstream targets of *SREBF1* (SREBP1) and *SREBF2* (SREBP2), TFs central to cholesterol biosynthesis (Fig. 3B,C; Appendix Fig. S3E,F). In line with this, gene expression of well-known SREBP targets, including *ACSL1, ACSS2, INSIG1,* and *FDPS,* reflected the expected transcriptional changes within the cholesterol pathway after PCZ-PIT combination treatment (Fig. 3C).

To further investigate the activity of SREBP TFs upon PCZ and/or PIT treatment, we analyzed protein expression and the presence of the cleaved (activated form) of both factors SREBP1 and 2 by western blot (WB). Significant SREBP1 full-length protein decrease occurred with PIT treatment, which is an expected effect of on-target statin activity (Fig. 3D,E). We also observed decreased levels of full-length (fl) SREBP2 in the PCZ + PIT group (Fig. 3D,E), suggesting that SREBP2 is activated when PCZ + PIT combination is present and cellular cholesterol levels become low. The subcellular localization of SREBP2 to the nucleus confirmed the presence of the active form of the SREBP2 (Fig. 3F,G). WB analysis also showed evidence of PARP-associated cell death but no statistically significant effect on MYCN protein expression after combination treatment when compared to controls (Fig. 3D,E). The stable MYCN protein levels after single PCZ

and PIT treatment further suggest that the mechanism-of-action is MYCN-independent (Fig. 3D,E).

## Dysregulation of cholesterol homeostasis by prochlorperazine-pitavastatin combination

We further explored the downstream effects of the two different single drugs and their combination. First, to confirm on-target effect of PIT we performed siRNA knockdown of *HMGCR* with and without PCZ treatment. HMGCR is the key rate-limiting enzyme in the mevalonate pathway, a critical step in cholesterol biosynthesis. Statins, including PIT, bind competitively to HMGCR, reducing de novo cholesterol production. Knockdown of *HMGCR* resulted in decreased NB cell viability without PCZ (siRNA #1) and in a concentration-dependent manner after addition of PCZ (siRNA#1 and #2) (Fig. 4A). WB analysis confirmed downregulation of HMGCR in the knockdown and an upregulation of cleaved (active) SREBP2 in both the knockdowns and after treatment with PCZ (Fig. 4B,C; Appendix Fig. S3G). Rescue experiments to reverse the effect of PIT with mevalonate, a product of the HMGCR reaction, confirmed the mechanism of statin-induced inhibition of the rate-limiting HMGCR enzyme in the mevalonate pathway (Fig. 4D).

However, mevalonate did not alter the viability of PCZ-treated cells, suggesting a different mechanism of action. (Fig. 4D). Phenothiazines (e.g., PCZ) can exert multiple effects that might alter cell proliferation and cell death, including inducing apoptosis, G1 cell cycle arrest, and membrane destabilization. Our transcriptomic data suggested that PCZ led to dysfunctional cholesterol metabolism (Fig. 3A; Appendix Fig. S3B). To investigate cellular cholesterol abundance and intracellular localization during treatment, we stained free cholesterol with filipin and observed an increased and more localized signal after combination treatment (Fig. 4E). Live cell experiments with LysoTracker and labeled cholesterol revealed colocalization and accumulation of cholesterol in lysosomes after PCZ and PCZ + PIT combination treatment but not as pronounced with PIT treatment alone (Fig. 4F). Together, our results suggest that PCZ leads to cholesterol accumulation in lysosomes and an overall cellular deficit in cholesterol.

Treatment-induced metabolic profiles were further investigated using lipidomic analysis, which revealed significant changes in individual lipid species (Fig. S4). Specifically, there was an increase in sphingomyelin lipid class after PCZ treatment which has previously been linked to phenothiazine's cytotoxic effects (Mehrabi et al, 2023) (Fig. 4G). Decrease in lysophosphatidylcholines after PIT treatment could point to decreased fatty acid pool

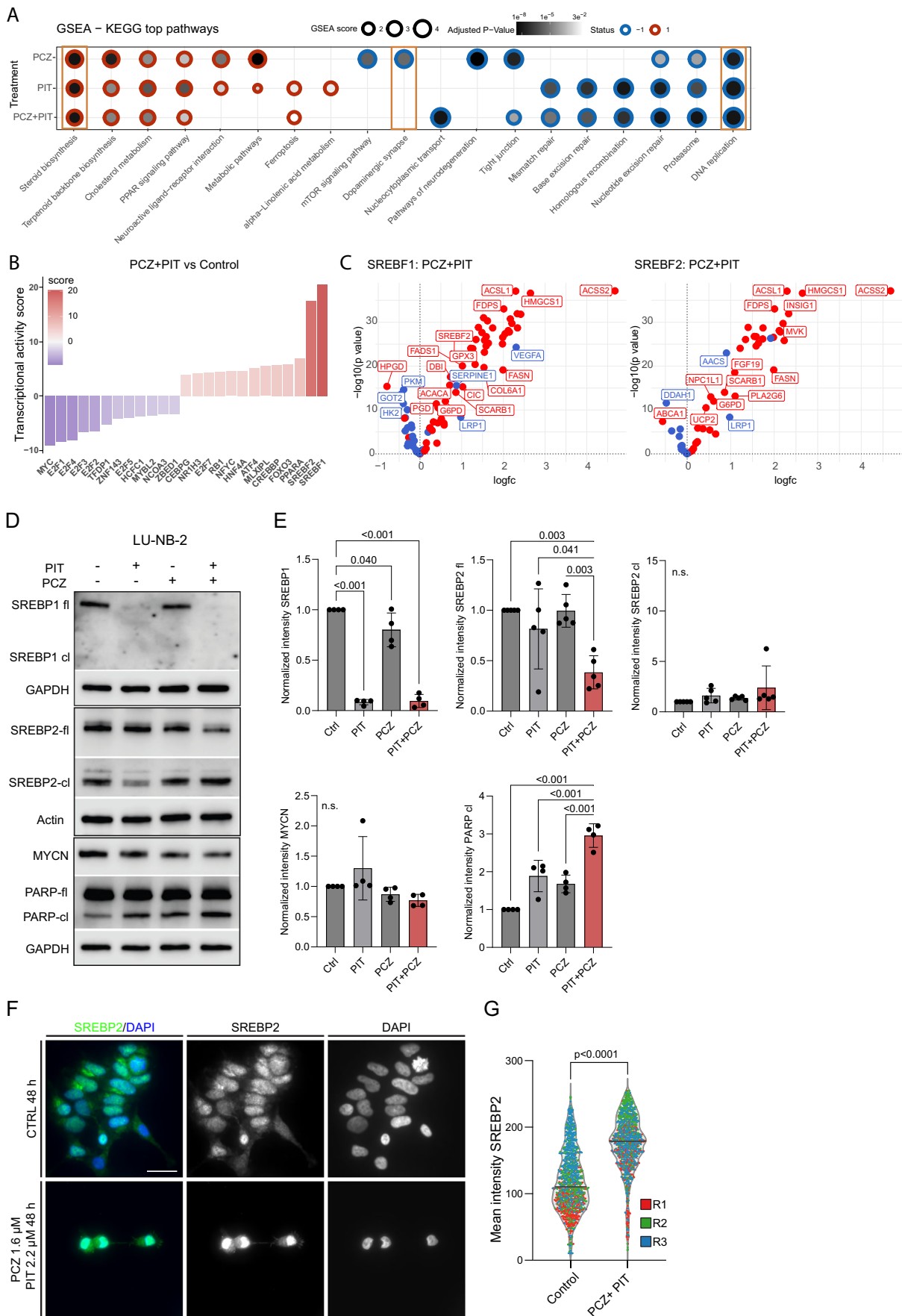

◄ **Figure 3. Dysregulated cholesterol metabolism after PCZ + PIT treatment.**

(A) GSEA transcriptomic analysis of treatment response in LU-NB-2 PDX cells after 48 h treatment using KEGG pathways, average of $n = 2*4$ per condition (biological*technical replicates). (B) Transcription factor (TF) activity inferred with CollecTRI after combination treatment. (C) Volcano plots displaying expression of the downstream targets of transcription factors SREBF1 and SREBF2 after PCZ + PIT combination treatment. (D) Western blot analysis of treatment response after 48 h. Samples within one biological replicate (experiment) were run in parallel to ensure the capture of multiple markers of similar size ($n = 4$-5 biological replicates). (E) Quantification of protein expression of SREBP1 ($n = 4$), SREBP2 and cleaved SREBP2 ($n = 5$ each), followed by MYCN and cleaved PARP (bottom row) ($n = 4$, one-way ANOVA followed by Tukey's multiple comparison test). (F) Images of immunofluorescence of SREBP2 in LU-NB-2 cells after 48 h PCZ-PIT combination treatment ($n = 3$ biological replicates). (G) Colocalization analysis of LU-NB-2 cells, mean intensity of SREBP2 fluorescence signal in the nucleus. Data presented as median violin plot; each dot represents a cell. For each condition, 382–432 DAPI+ cells were counted (i.e., three biological replicates (R)* two conditions: control and PIT 2.2 µM + PCZ 1.6 µM treated cells). Kolmogorov–Smirnov analysis for testing the assumption of normality, followed by two-way ANOVA and Bonferroni´s nonparametric test for data with non-normal distribution. Scale bar 20 µm. GSEA gene set enrichment analysis, PCZ prochlorperazine, PIT pitavastatin, TF transcription factor. Source data are available online for this figure.

needed for cell proliferation (Fig. 4G) (Kamphorst et al, 2013; Broadfield et al, 2021). These changes are consistent with the known effects of the drugs influence on lipid and cholesterol availability and function in cancer cells (Broadfield et al, 2021).

Thus, PIT-induced inhibition of de novo cholesterol synthesis combined with PCZ-induced cholesterol accumulation in lysosomes creates a double-strike on cholesterol availability in NB.

## Combined prochlorperazine and pitavastatin leads to a chemosensitive NB cell state in vitro

We next investigated the effect of the PCZ + PIT combination on NB cell states in vitro. Given that different gene sets can identify ADR and undifferentiated MES–like cell states, we explored multiple signatures derived from in vitro models, xenografts, and patient tumors (van Groningen (Van Groningen et al, 2017), Boeva (Boeva et al, 2017), Gartlgruber (Gartlgruber et al, 2021), Olsen (Olsen et al, 2024), Manas (Mañas et al, 2022), Bedoya-Reina (Bedoya-Reina et al, 2021), Yuan (Yuan et al, 2022), Thirant (Thirant et al, 2023), and Patel (Patel et al, 2024); Appendix Table S4). GSEA showed enrichment of several ADR gene signatures derived from patient tumors (Bedoya-Reina, Yuan, and Patel) and an integrated signature from multiple datasets (Manas) (Fig. 5A,B). Further, treatment decreased one undifferentiated MES-like signature (Manas) (Fig. 5A,C). The decrease in the recently described transitional subtype of aggressive bridging cells (Yuan) (Fig. 5C), as well as the strong decrease in SYMP-Patel signature (defining actively proliferating NB cells of mainly ADR subtype) (Fig. 5A), likely reflects decreased NB cell proliferation following treatment.

Considering that the original predictions were based on features of high-risk vs low-risk tumors, we further investigated the relationship between PCZ + PIT combination-induced expression profile and risk features of NB patient tumors in two independent datasets. Using publicly available NB datasets (SEQC 498 and 161 Target), we found that genes upregulated after PCZ + PIT treatment were upregulated in low-risk NB tumors, whereas treatment-induced downregulated genes were representative of high-risk NB tumors (Fig. 5D; Appendix Fig. S5A,B). In addition, we analyzed the datasets originally included in the in silico predictions (Appendix Table S1) and concluded that the transcriptional responses to PCZ and/or PIT were indeed associated with lower risk/stage groups (Appendix Fig. S5C).

Based on the drug-induced increase in an ADR signature, we hypothesized that pretreatment with the drug combination would sensitize NB cells to chemotherapy. Thus, we tested the combined PCZ + PIT treatment together with cisplatin in treatment-resistant NB organoids, LU-NB-1 and LU-NB-2. As predicted, different treatment sequences revealed that pretreatment with combined PCZ + PIT followed by cisplatin resulted in the lowest NB cell viability (Fig. 5E; Appendix Fig. S5D), suggesting that PCZ-PIT combination pretreatment can sensitize chemoresistant NB to chemotherapy.

Together, our data suggest that PCZ + PIT treatment leads to a phenotypic change towards an ADR cell state in vitro and that the treatment-induced response correlates with transcriptional profiles of low-risk NB. Importantly, PCZ + PIT sensitizes chemoresistant NB to chemotherapy in vitro.

## Combined prochlorperazine and pitavastatin sensitizes chemoresistant NB to chemotherapy in vivo

Encouraged by the in vitro findings, we evaluated the effect of combined PCZ + PIT with standard-of-care chemotherapy in vivo. We used the chemoresistant LU-NB-1 PDX model, which displays a very limited response to COJEC induction therapy (Mañas et al, 2022).

In a short-term in vivo study for 10 days (Fig. 6A), both the drug combination and the combination + COJEC inhibited NB tumor growth ($P = 0.0045$ and $P < 0.0001$, respectively; mixed effects analysis with Dunnett's multiple comparison test), with the triple combination being clearly more effective (Fig. 6B). The tumors had a mean starting volume of ~300 mm³, and no toxicity and weight loss occurred over the duration of the study (Fig. EV2A,B). To investigate the mechanisms of treatment response at the transcriptional level, we performed RNA-seq of tumors after 10 days of treatment. This analysis revealed upregulation of diverse stress responses and cholesterol homeostasis and downregulation of oxidative phosphorylation and MYC targets in the treated groups (Fig. 6C).

Transcriptional changes included upregulation of signatures defining both the ADR and MES-like cell states (Fig. 6D). Thus, while PCZ + PIT treatment resulted in an increased ADR phenotype in vitro (Fig. 5A,B), in vivo treatment enhanced both ADR and MES phenotypic features. Multiplex immunofluorescence staining using selected proxy markers for NB cell states (NCAM (NB); PHOX2B (ADR), TH (ADR) and SOX9 (MES)) revealed heterogenous response and an upregulation of the proportion of PHOX2B positive NB cells after treatment with the PCZ + PIT combination as well as with combination +

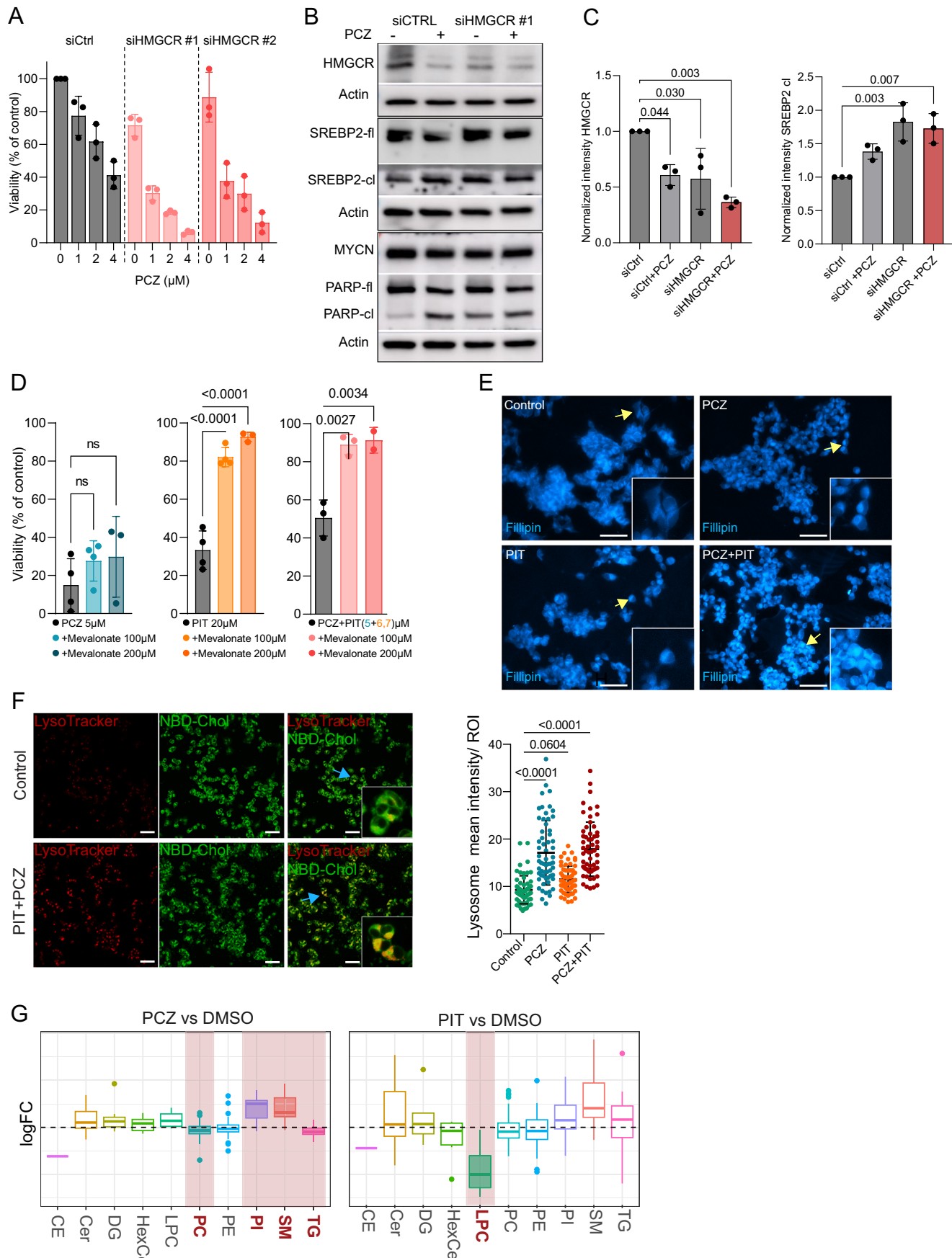

◀ **Figure 4.  Downstream effects of PCZ and PIT.**

(A) Viability of LU-NB-2 organoids after siRNA knockdown of HMGCR by two different siRNA probes (#1 and #2), (mean percent viability of control ±SD, $n = 3$). (B) Western blot (WB) analysis after siRNA knockdown of HMGCR (#1) with and without PCZ treatment for 48 h. Samples within one biological replicate (experiment) were run in parallel to ensure the capture of multiple markers of similar size ($n = 3$). (C) Quantification of HMGCR (left) and cleaved SREBP2 in WB, mean normalized intensity ±SD, $n = 3$ (one-way ANOVA with Tukey's multiple comparisons test). (D) Mevalonate rescue experiments after treatment with PCZ, PIT, and combination in LU-NB-2 organoids, mean percent viability of control ±SD, $n = 4$ (PCZ or PIT), $n = 3$ (PCZ + PIT) (one-way ANOVA followed by Dunnet's multiple comparisons). (E) Representative photos of treated LU-NB-2 cells stained with the free-cholesterol binding dye filipin (scale bar 50 µm, yellow arrows indicate magnified region). (F) Live cell imaging using LysoTracker staining and labeled cholesterol (scale bar 50 µm) followed by quantification of mean lysosome intensity in 20 selected ROIs (cells); 24 h, $n = 3$ (one-way ANOVA with Tukey's multiple comparisons test, lines represent mean and SD values). (G) Lipidomics analysis showing changes in abundance of lipid classes in PCZ and PIT treatment groups compared with DMSO-treated controls; significant changes marked in red (permutation test was used to calculate $P$ values and corrected with Benjamini–Hochberg test, adj.$P$ values: PC:0.017, PI:0.023, SM:0.001, TG:0.006, LPC:0.016), $n = 6$. PCZ prochlorperazine; boxes represent the interquartile range (IQR), center lines indicate medians, whiskers extend to 1.5×IQR, and points denote outliers). PIT pitavastatin, LDL low density lipoprotein, LDLR low density lipoprotein receptor, CE cholesterol esters, Cer ceramides, DG diacylglycerol, HexCer hexosylceramide, LPC lysophosphatidylcholine, PC phosphatidylcholine, PE phosphatidylethanolamine, PI phosphatidylinositol, SM sphingomyelin, TG triglycerides. boxes represent mean values and whiskers indicate SD. Source data are available online for this figure.

chemotherapy (PhenoImager HT 2.0, Akoya, Figs. 6E and EV2C). Spatial analysis at the single cell level revealed co-expression of two or three of the markers in all treatment groups (Figs. 6F and EV2C), indicating that these markers can be co-expressed in the same NB cell.

Finally, we compared PCZ-PIT combination + COJEC and COJEC treatment alone in a long-term survival study of chemoresistant LU-NB-1 xenografts. Mice were randomized into four groups, and the treatment was initiated when tumors reached 200–300 mm³ (Figs. 6G and EV2D). All tumors that reached that size were pre-treated with two i.t. injections of the combination before the i.p. COJEC treatment started (Fig. 6G). Intratumoral drug injections were performed according to a standardized schedule to ensure good drug distribution into the tumor (Fig. EV2E). The addition of PCZ-PIT did not increase the side-effects expected from COJEC chemotherapy (Fig. EV2F). Mice were monitored for toxicity and showed <10% weight loss from baseline in the treatment groups that included chemotherapy (Fig. EV2F).

Treatment with PCZ + PIT combination + COJEC decreased tumor growth compared with controls at day 17 ($P = 0.0394$; Tukey's multiple comparison test of all groups; the last day of all mice alive) (Figs. 6H,I and EV2G). Tumor regression was observed in one control mouse, likely caused by the i.t. injections. A fraction of mice responded to combination or COJEC treatment alone, but tumors rapidly progressed after treatment was stopped at day 30 (Fig. EV2G). Tumors in three of the six COJEC-treated mice initially regressed to 0 mm³ but quickly relapsed after treatment removal, with a median group survival of 62.5 days (range 19–96 days; Figs. 6J and EV2G). In the combination + COJEC group, six out of seven tumors initially regressed completely (Figs. 6H and EV2G), and survival was extended to a median of 85 days (range 52–210 days; Fig. 6J). One mouse in the combination + COJEC group did not relapse throughout the study period (210 days). Importantly, survival was significantly prolonged in the combination + COJEC group compared with both the control ($P = 0.006$) and COJEC-treated mice ($P = 0.027$, two-sided log-rank tests; Fig. 6J). This result underscores the benefit of adding the combination to standard-of-care chemotherapy.

In summary, the addition of PCZ + PIT combined therapy to standard-of-care COJEC chemotherapy outperforms COJEC chemotherapy alone and significantly prolongs survival in a chemoresistant NB PDX model.

## Discussion

Chemoresistant high-risk NB is a major challenge in pediatric oncology, and novel treatment strategies are warranted. Through a drug repurposing approach, we identified a novel combination of statins and phenothiazines with strong anti-NB properties. Exploiting DGEM based on multiple gene expression profiles in combination with the PRISM drug prediction tool, we identified effective and safe new therapies for high-risk NB. The novel drug combination was synergistic, altering cholesterol metabolism through two distinct yet converging disruptions leading to cholesterol deficiency. This dual interference (double-strike) also resulted in an ADR-like chemosensitive NB phenotype in vitro. Treatment with a combination of PCZ + PIT together with standard-of-care COJEC chemotherapy outperformed chemotherapy alone and prolonged survival in a chemoresistant *MYCN*-amplified PDX model.

Drug repurposing has the potential to accelerate the long and expensive path from drug discovery to a clinically useful medication. By using already available safety, efficacy, and pharmacokinetic data, this approach is of particular interest for childhood and rare diseases where clinical trials are difficult to perform (Pushpakom et al, 2019; Schipper et al, 2022). Diverse strategies have been applied to identify drugs for repurposing, including the Drug Repurposing Hub at Broad Institute (Corsello et al, 2017), TargetTranslator (Almstedt et al, 2020), and DGEM (Tranfaglia et al, 2019). In this study we used two tools to predict potential treatments; the PRISM "guilt by association" tool, and the DGEM gene expression matching tool. Between these, many thousands of comparisons between diseases and compounds were carried out to arrive at a set of compound treatment predictions.

Twelve drugs were identified in silico for repurposing in high-risk NB, and subsequent experiments narrowed the list down to two classes of therapeutics: cholesterol synthesis inhibitors (statins) and antipsychotic dopamine receptor (D2) antagonists (phenothiazines). Statins inhibit HMG-CoA reductase which reduces cholesterol production via inhibition of the mevalonate pathway. Statins are known for reducing the risk of cardiovascular disease; however, the impact of these drugs on cancer has been investigated for decades, and experimental studies in NB suggest that statins can induce apoptosis and differentiation (Almstedt et al, 2020; Girgert et al, 1999; Arnold et al, 2010; Marcuzzi et al, 2012). The

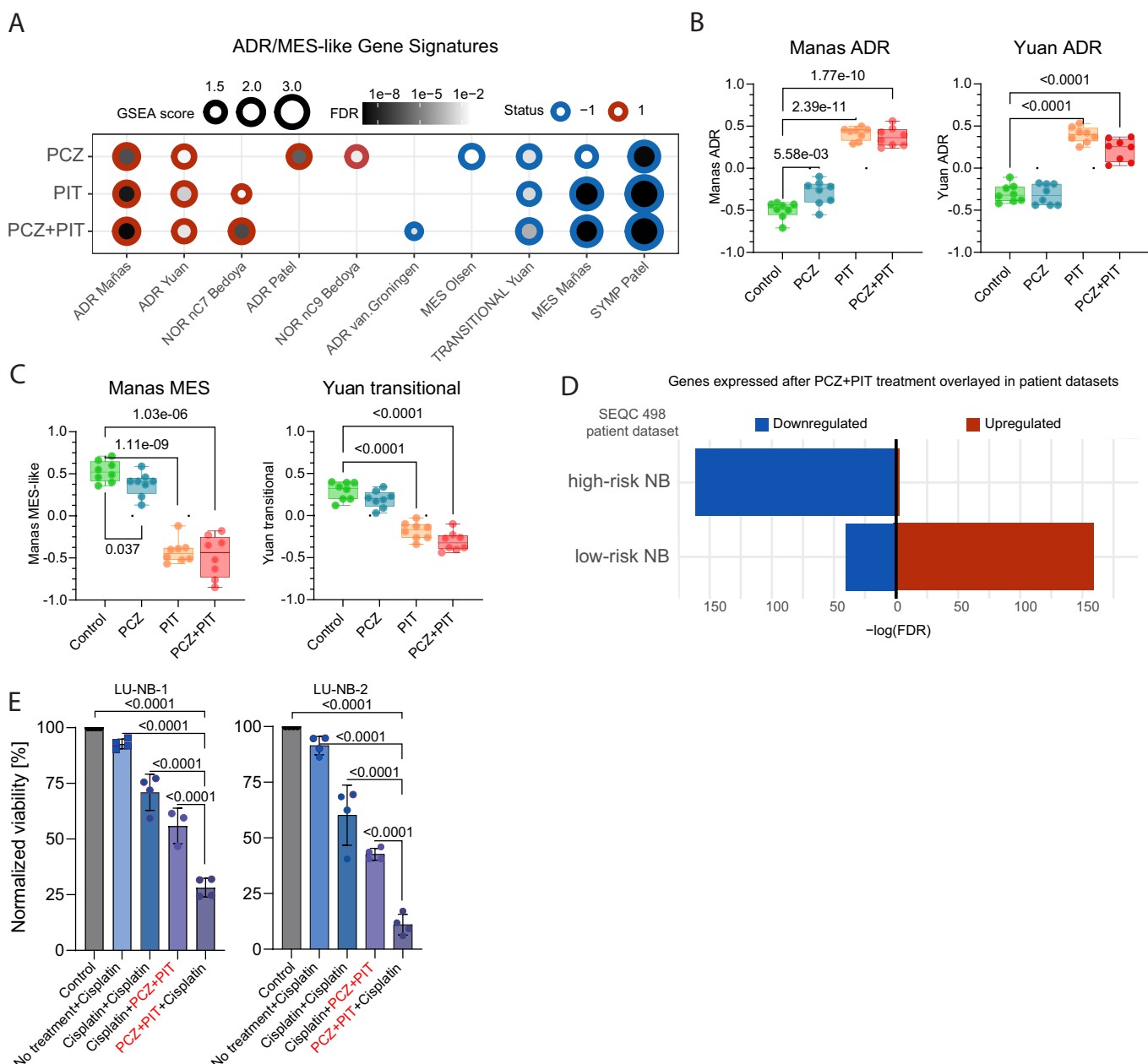

**Figure 5. Phenotypic change after PCZ + PIT treatment in vitro.**

(A) Association between various MES/ADR-like gene signatures and DEGs in LU-NB-2 organoids treated with PCZ and/or PIT for 48 h. Only significant (FDR *P* value < 0.05) results are displayed, average of *n* = 2*4 per condition (biological*technical replicates). (B) Comparison between treatment groups for two individual ADR-like gene sets (*n* = 8 in each group, Welch *t* test, boxes represent the interquartile range and whiskers indicate minimum and maximum values). (C) Comparison between treatment groups for the Manas MES-like and Yuan transitional gene signatures (*n* = 8 in each group, Welch *t* test, boxes represent the interquartile range and whiskers indicate minimum and maximum values). (D) Association of treatment-induced gene expression with high-risk and low-risk NB features obtained from the public patient dataset SEQC498. Genes downregulated by PCZ + PIT treatment are enriched in high-risk neuroblastoma patients. (E) NB cell viability after various sequential treatment strategies of the PCZ + PIT drug combination (1.6 μM + 6.7 μM) and cisplatin (5 μM), *n* = 4 (one-way ANOVA with Tukey's multiple comparisons test). Boxes represent mean values, and whiskers indicate SD. PCZ prochlorperazine, PIT pitavastatin, Combination PCZ + PIT, MES like-undifferentiated mesenchymal-like cells, ADR–adrenergic/noradrenergic-like cells. Source data are available online for this figure.

antipsychotic phenothiazines, e.g., PCZ, are also indicated for use against nausea and migraine during chemotherapy treatment (Varga et al, 2017). Here, we identified several phenothiazines (TFP, TRD, and PCZ) with potential anti-NB effects, and we selected PCZ as the most promising agent. Importantly, combined

PCZ + PIT therapy yielded substantial additive or synergistic effects in high-risk NB.

As expected, gene expression and molecular analyses suggested that the mechanism of action of PIT was related to the mevalonate pathway (Juarez and Fruman, 2021). Consequently, the inhibitory

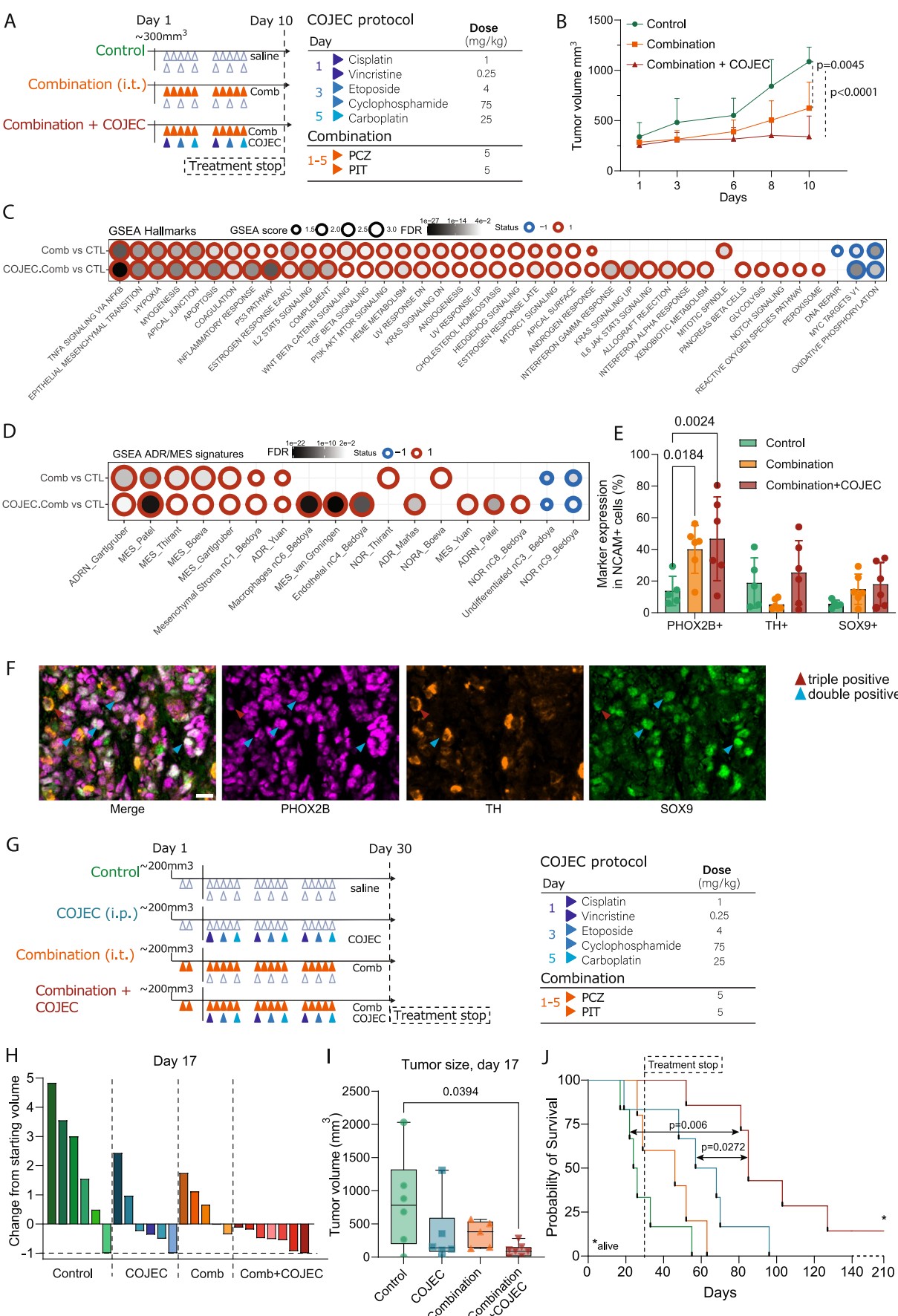

◄ **Figure 6. PCZ + PIT combination enhances standard-of-care COJEC compared to COJEC alone.**

(A) Schematic overview of short-term in vivo study of the combination PCZ + PIT (i.t.) with COJEC (i.p.). Mice with LU-NB-1 tumors were randomly allocated to the groups and treated with the corresponding schedules for 10 days. (B) Change in tumor volume over time for all groups, control = 7, combination = 7, combination + COJEC = 6, (mixed effects analysis with Dunnett's multiple comparison test). (C) RNA-seq and gene set enrichment analysis of 20 tumors. Significant contrasts in expression between the two treatment groups and the control are displayed; Hallmarks database. (D) Enrichment of MES- and ADR-like gene signatures after PCZ + PIT treatment in vivo. (E) Expression of PHOX2B (ADR), TH (ADR), and SOX9 (MES) in NCAM+ (NB) cells (%). Image analysis of multiplex immune fluorescent (IF) staining (PhenoImager HT 2.0, Akoya Biosciences), n = 2 tumors from each group*3 areas per tumor (two-way ANOVA with Bonferroni's nonparametric test). (F) Representative images of single marker expression and of co-expression (arrows) after PCZ + PIT combination treatment. PHOX2B = magenta, TH = orange, SOX9 = green, co-expression of PHOX2B and SOX9 emerge as white. Scale bar = 20 μm. (G) Schematic overview of the long-term survival study in mice bearing LU-NB-1 tumors. Two doses of the combination were given before the start of standard-of-care COJEC chemotherapy. Treatment was given for 30 days. (H) Day 17 (last day of all mice alive) single mouse results showing the ratio of tumor size change from baseline control = 6, COJEC = 6, combination = 5, combination + COJEC = 7. (I) Summary of day 17 tumor volume (one-way ANOVA followed by Tukey's multiple comparison test of all groups; boxes represent the interquartile range and whiskers indicate minimum and maximum values.). (J) Kaplan–Meier survival graph showing survival of mice treated with combination + COJEC compared with control (log-rank test P = 0.006) or standard-of-care (log-rank test P = 0.0272). PCZ prochlorperazine, PIT pitavastatin, combination PCZ + PIT, i.t. intratumoral, GSEA gene set enrichment analysis, MES like-undifferentiated mesenchymal-like cells, ADR adrenergic/noradrenergic-like cells. Source data are available online for this figure.

effect on viability was rescued by mevalonate. Conversely, even if we cannot exclude that downregulation of dopaminergic activity is of partial importance for the antitumor effect of PCZ, this mechanism does not seem to be the main mechanism of action. Instead, the dominant pattern of treatment response after PCZ was strong upregulation of cholesterol biosynthesis and metabolism, similar to recent findings described in glioblastoma after treatment with phenothiazines (Bhat et al, 2021). Cholesterol metabolism is known to be dysregulated in cancer, since both synthesis and uptake are often increased to meet the metabolic needs of cancer cells (Abdulla et al, 2021). While statins are known to inhibit de novo cholesterol production, our data from live cell imaging and IF staining showed accumulation of cholesterol and colocalization of cholesterol and lysosomes after treatment with combined PCZ + PIT. Thus, our data suggests that PCZ + PIT targets cholesterol in cancer through a double-strike mechanism of cholesterol accumulation in lysosomes (PCZ) and inhibition of de novo cholesterol synthesis (PIT), leading to SREBF2 activation and translocation to the nucleus. This finding is consistent with recent studies suggesting that phenothiazines can inhibit lysosome function, leading to impaired intracellular delivery of LDL-derived cholesterol (Xia et al, 2021). It has been suggested that phenothiazine-induced impairment of lysosome function can block lysosomes and autophagosome fusion, also leading to autophagy inhibition (Lopes et al, 2024). In turn, dysfunctional autophagy can lead to the accumulation of toxic products such as ROS and oncogenic proteins in autophagosomes, causing the eventual death of NB cells (Li et al, 2020).

The PCZ + PIT combination also led to an ADR phenotype in vitro. Given previous reports implicating MES-like NB cells in chemotherapy resistance (Van Groningen et al, 2017; Mañas et al, 2022), we hypothesized that the drug-induced phenotypic change might enhance NB chemosensitivity. Indeed, using a chemoresistant PDX-model and a clinically relevant chemotherapy treatment protocol (Mañas et al, 2022), we showed that mice pre-treated with the PCZ + PIT combination (two days) and thereafter exposed to both PCZ + PIT and standard-of-care chemotherapy displayed longer survival compared with standard-of-care chemotherapy alone. Interestingly, transcriptional analysis pointed to upregulation of both ADR and MES cell states after treatment in vivo. Although we did not perform single-cell RNA-sequencing, our findings indicate context-dependent (in vitro vs. in vivo) differential responses of NB cell

states. Furthermore, multiplex staining revealed single-cell co-expression of proxy markers for the ADR and MES states, suggesting that these states may not be mutually exclusive. However, lineage tracing, single-cell analyses, and additional cell state markers are required to confirm this.

Recent findings in small cell lung cancer (SCLC) and pancreatic cancer have shown that statin treatment can sensitize tumors to chemotherapy to produce durable responses in a pilot study of patients with relapsed SCLC after chemotherapy (Guo et al, 2022), supporting this approach to treat chemoresistant and relapsed cancer. Similarly, statins can enhance chemotherapy efficacy in pancreatic ductal adenocarcinoma (PDAC), with 70% of patients showing a response in a phase 2 clinical trial (Li et al, 2025). Suggested molecular mechanisms explaining increased chemosensitivity upon cholesterol deregulation include destroyed lipid rafts leading to aborted multi-drug resistant (MDR) ABC transporter activity (Liu et al, 2021; Glodkowska-Mrowka et al, 2014; Wu et al, 2015; Gupta et al, 2018) as well as downregulation of pathways of importance for epithelial-to-mesenchymal (EMT) cell state transition (e.g., Wnt, TGF-β) (Abdulla et al, 2021). The latter relationship between cholesterol homeostasis and cell state changes could (at least partially) explain the observed phenotypic change in NB upon PCZ + PIT targeting, but further studies will be necessary to confirm causation.

This study has limitations. The mechanistic analyses were primarily conducted in MYCN-amplified NB models, which may limit the generalizability of our findings to all high-risk NBs. However, the drug prediction analyses were based on patient datasets encompassing both MYCN-amplified and non-amplified NB, indicating potential broader applicability. Supporting this, we observed reduced NB cell viability following single-agent and PCZ + PIT combination treatment in two non-MYCN-amplified cell lines. Moreover, treatment with PCZ and/or PIT did not alter MYCN protein levels in MYCN–amplified models, suggesting that MYCN is not a key mediator of the observed drug responses.

A challenging aspect of the novel combination of statins and phenothiazines is the bioavailability of the drugs in oral form after first-pass metabolism in the gut and liver (Juarez and Fruman, 2021; Björkhem-Bergman et al, 2011; Finn et al, 2005). Consequently, intraperitoneal injection of the drugs did not produce significant effects as statins can accumulate in the liver despite bypassing the gut metabolism. Pitavastatin, despite having the highest bioavailability and different metabolic fate than other

statins, is still preferentially distributed to the liver and intestines, and it is likely the reason for the drug to not circulate at biologically relevant concentrations (Fujino et al, 1998). Despite evidence that mevalonate and cholesterol are important in cancer (Liu et al, 2021), no selective targeting drugs are yet available. In the present study, anti-NB effects were obtained by intratumoral injection, which would be challenging in a clinical setting in patients with multiple distant metastases, so a new formulation might be necessary to improve tumor exposure. Alternatively, tumor-specific targeting of the suggested combination might be possible. Antibodies and CAR-T cells targeted against disialoganglioside (GD2) are already used in clinical practice against high-risk NB today. Importantly, the safety of novel combinations within a new target group, e.g., children, must also be considered afresh (Schipper et al, 2022; Lau Moon Lin et al, 2016). Nevertheless, there are successful examples of repurposing adult drugs for pediatric patients in other diseases; for example, fenfluramine, developed in the 60 s as an appetite suppressant, has been shown to be effective in Dravet syndrome, a rare neurological disease of infancy, and the FDA approved fenfluramine for this use in 2020 (Lagae et al, 2019; McCay et al, 2025).

The optimal sequence, combination, and dosing of PCZ + PIT with standard-of-care remains to be defined. Here, we demonstrated that pre-treatment with PCZ + PIT enhanced chemotherapy sensitivity in vitro. In vivo, a two-day pre-treatment with the drug combination prior to chemotherapy resulted in tumor regression and extended survival. However, further studies are needed to establish the most effective treatment schedule, particularly using clinically approved formulations of the compounds. Pitavastatin and other statins are currently being investigated in clinical trials to assess their bioavailability in glioblastoma (NCT05977738) and their effects as part of combination treatment in patients with leukemia (NCT04512105) and breast cancer (NCT03358017). Prochlorperazine is FDA-approved for indications including nausea and vomiting in a post-chemotherapy, post-operative setting, and off-label use to treat acute migraines in adults and children. These antiemetic effects could be advantageous in the treatment of children with cancer taking chemotherapy (Bachur et al, 2015; Sheridan et al, 2018; Din and Preuss, 2024). Recent findings suggest that PCZ can also enhance the anti-tumor effects of monoclonal antibody therapies, possibly through inhibition of endocytosis (Chew et al, 2020). Consequently, there are ongoing clinical trials of high doses of PCZ in combination with paclitaxel, trastuzumab, and pertuzumab for HER2-positive metastatic cancer (Downton et al, 2023).

In summary, using in silico drug predictions and experimental drug testing in patient-derived tumor models, we show that transcriptomics and machine learning based drug–disease matching is a viable option to identify new treatment strategies against NB. We identified two classes of drugs that, when administered together, hold promise for patients with chemoresistant NB. With increasing numbers of single-cell transcriptional profiles of aggressive, resistant tumors and rapidly developing deep learning methods, we expect that this drug repurposing approach will yield additional promising therapeutic strategies against treatment-resistant cancers.

# Methods

## Reagents and tools table

| Reagent/resource | Reference or source | Identifier or catalog number |
|---|---|---|
| **Experimental models** | | |
| SK-N-SH | ATCC | HTB-11 |
| SK-N-AS | ATCC | CRL-2137 |
| NOD scid gamma (NSG) mice | Taconic | RRID:IMSR_JAX:005557 |
| **Antibodies** | | |
| pAktSer | Cell Signaling Technology | 193H12 |
| panAkt | Cell Signaling Technology | 40D4 |
| MYCN | Cell Signaling Technology | 51705 |
| SREBP2 | Abcam | AB30682 |
| SREBP1 | Cell Signaling Technology | 95879 |
| HMGCR | Thermo Fisher Scientific | MA5-31335 |
| PARP | Cell Signaling Technology | 9542 |
| GAPDH | R&D Systems | 2275-PC-100 |
| Actin | Thermo Fisher Scientific | MA5-15739-HRP |
| DAPI | Invitrogen | D3571 |
| Filipin | Abcam | ab133116 |
| LysoTracker™ Red DND-99 | Invitrogen | L7528 |
| NBD-labeled free cholesterol | Abcam | ab269448 |
| NCAM | DAKO | NCL-L-CD56-504 |
| SOX9 | Abcam | Ab76997 |
| TH | Abcam | Ab112 |
| CD44 | Abcam | Ab157107 |
| PHOX2B | Abcam | Ab183741 |
| **Oligonucleotides and other sequence-based reagents** | | |
| siRNAs targeting *HMGCR* #1 | Ambion, Life Technologies | ID s142; 5′-UAUAAUCCCUGUAAGGUAGgt-3′ |
| siRNAs targeting *HMGCR* #2 | Ambion, Life Technologies | D s143; 5′-GGUUCGCAGUGAUAAAGGAtt-3′ |
| **Chemicals, enzymes, and other reagents** | | |
| Propidium iodide (1 mg/ml) | Invitrogen | P3566 |
| Mevalonolactone | Sigma-Aldrich | M4667 |
| trifluoperazine | Selleckchem | S3201 |
| thioridazine | Selleckchem | S5563 |
| nortriptyline | Selleckchem | S3698 |

| Reagent/resource | Reference or source | Identifier or catalog number |
|---|---|---|
| prochlorperazine | Selleckchem | S4631 |
| resveratrol | Selleckchem | S1396 |
| sildenafil | Selleckchem | S1431 |
| dasatinib | Selleckchem | S1021 |
| pazopanib | Selleckchem | S3012 |
| lovastatin | Selleckchem | S2061 |
| folic acid | Selleckchem | S4605 |
| topiramate | Selleckchem | S1438 |
| tianeptine | Selleckchem | S5087 |
| fluvastatin | Selleckchem | S1909 |
| pitavastatin | Selleckchem | S1759 |
| Software | | |
| SynergyFinder | FIMM | https://synergyfinder.fimm.fi |
| SynToxProfiler | FIMM | https://syntoxprofiler.fimm.fi |
| Zen 3.1 | Carl Zeiss | RRID:SCR_013672 |
| QuPath 0.4.3/0.5.0 | open source | RRID:SCR_018257 |
| FlowJo v10.1.0 | BD Biosciences | RRID:SCR_008520 |
| InForm | Akoya Biosciences | 3.0 |

## Experimental design

The objective of this study was to identify and test approved drugs for repurposing in high-risk neuroblastoma. We used a combination of in silico prediction methods (disease-gene expression matching and machine learning), in vitro experiments with neuroblastoma PDX-derived organoids, and in vivo high-risk neuroblastoma PDX models to investigate the effects of these drugs. For in vitro experiments, two or three biological replicates were performed (reported for each experiment). All in vivo studies were conducted in compliance with the guidelines of the Regional Ethics Committee for Animal Research in Lund (ethical permit no. 19012-19). Only female mice were used due to ethical considerations for long-term housing of adult males. Four- to eight-week-old female NOD scid gamma (NSG) mice (RRID:IMSR_JAX:005557) were purchased from Taconic. Animals' general health conditions were checked daily by professional caretakers, and housing facilities comply with local and European regulations. Female mice were housed in groups of up to five per standard individually ventilated cage (IVC) with bedding and nesting material as well as environmental enrichment provided. Animals were maintained under controlled environmental conditions, including a 12-h light/dark cycle, regulated temperature and humidity, and had ad libitum access to food and water. All procedures were conducted with efforts to minimize animal suffering and to reduce the number of animals used. For in vivo studies, dissociated PDX organoids were injected into NOD scid gamma (NSG) mice, and upon tumor engraftment, mice were randomly assigned to control or treatment groups with a minimum of five mice per group. Mice were euthanized on the basis of tumor size, weight loss, overall health deterioration, or end of study time. Tumor tissues were collected

from all mice for further analysis. All results were subjected to statistical analysis, with significance defined as $P < 0.05$. Investigators were not blinded to the experimental conditions, and no animals or data points in experiments were excluded from the analysis.

## In silico analyses for drug repurposing

Publicly available raw gene expression datasets were manually curated and analyzed for their suitability as input into Healx's predictive technologies. Four raw gene expression datasets passed quality control criteria that ensured internal robustness of the input data (Appendix Table S1). Gene expression datasets were derived from patient samples at various stages of progression of tumor growth, including those who had died of the disease. NCBI GEO datasets were used to derive the NB disease gene expression signatures.

## Differential gene expression profiles (GEPs) and pathway enrichment analysis

Neuroblastoma GEPs were calculated based on differentially expressed genes between groups of progressive/severe outcome and favorable outcome (details in Appendix Table S1). To avoid biasing DGEM compound prediction to any one specific subset of NB disease biology, we include a variety of conditions in the generation of gene expression query signatures: low-risk vs high-risk disease, *MYCN*-amplified vs non-*MYCN*-amplified tumors, and samples from patients with different clinical outcomes (alive vs diseased). This approach ensured comprehensive representation of NB biological heterogeneity in drug matching analyses, thereby enhancing the potential to identify compounds that target diverse and potentially novel biological mechanisms. Briefly, raw microarray datasets were downloaded from NCBI GEO, normalized by MAS5 or RMA, and subject to arrayQualityMetrics as defined by the curated contrasts to identify outliers. Datasets were preprocessed. Outlier samples were removed from the contrasts, and differential gene expression analysis was performed using Limma (RRID:SCR_010943). Adjusted $P$ values were calculated according to the Benjamini–Hochberg procedure to test for false discovery, after removing insignificant (adj.$P$ val > 0.05) and control probe sets from the signature. Gene expression profiles (GEPs) formed from differentially expressed genes in these datasets were evaluated with gene set enrichment analysis (GSEA) to ensure they described biological processes indicative of an NB phenotype (Appendix Fig. S1A).

## Drug matching

### Disease-gene expression matching (DGEM)
All GEPs passed our technical and biological quality assessments and were used to predict drugs with DGEM. Healx's DGEM implements analysis algorithms that automatically select the set of conditions most likely to yield informative drug–disease connections. Briefly, known NB therapeutic agents serve as positive controls and are used to assess the performance of the DGEM experiment. Prior approved NB therapies or agents in late-stage clinical trials are assumed to be highly ranked in the experiment. The degree to which these positive control compounds are enriched provides a mechanism to optimize DGEM parameters and rank the prediction sets generated.

### PRISM analysis

PRISM (RRID:SCR_005375) is a predictive methodology that utilizes known information about compound and disease similarities to infer novel associations between nearest-neighbor compounds and diseases. In detail, PRISM calculates high-dimensional feature vectors for each compound-disease pair. Similarly, to the repurposing method PreDR (Wang et al, 2013), PRISM employs a support vector machine (SVM) to identify clusters of known indication pairs within this high-dimensional space. We enhance the recall of this method by using a Kronecker product kernel in the SVM (Wang and Zhang, 2013). Pairs of compounds and diseases found within or near these learned clusters are predicted as potential novel treatment indications. For instance, suppose compound B is known to treat disease A (Fig. 1). Compound D has a similar structure to compound B so, by association, is a predicted treatment for Disease C. Similarly, Disease C (lower right; red) is associated with disease A by phenotype. Hence, compound B and compound D may be appropriate treatments for disease C.

### Literature mining

Healx employs literature mining and natural language processing to find compound/disease relationships in the scientific literature. Text mining was employed to identify mentions of compounds and diseases in the titles and abstracts of millions of scientific articles published from 2000 to 2016, and extracted sentences where both a compound and a disease occurred. We used deep learning methods to classify those sentences and generate therapeutic scores, predicting the likelihood that the compound could treat the disease. We previously validated this method and showed strong recall of known therapeutic relationships (Tranfaglia et al, 2019).

The evidence from this approach was automatically attached to predictions generated by other repurposing methods. Literature mining results were used as direct evidence for an association between a drug and a disease but also to infer evidence from related diseases and drugs. Literature mining is an important addition to known drug indications, as it can capture off-label treatment and preclinical experimental results.

### Treatment profiles

Treatment profiles specify the properties and medicinal qualities of a prospective drug. Healx referenced the NB treatment profile during the derivation of the recommended predictions.

## Drug repurposing recommendations

Healx performed an expert pharmacological review of the top-ranking drug predictions across each of the GEPs that were used as input to DGEM. Prediction results were first analyzed by domain experts at Healx, who provided annotations with supporting pharmacodynamic data from an internal knowledgebase relevant to inferring therapeutic benefit in NB, such as literature support, clinical trials, and drug mechanism of action. This produced a ranking and filtering of strongly predicted drugs that were likely to be most efficacious, such that a case may be made for prospective validation of each drug by a team of drug repurposing experts.

## Drugs

Fourteen small molecule drugs were purchased from Selleckchem (Houston, TX): trifluoperazine (S3201), thioridazine (S5563),

nortriptyline (S3698), prochlorperazine (S4631), resveratrol (S1396), sildenafil (S1431), dasatinib (S1021), pazopanib (S3012), lovastatin (S2061), folic acid (S4605), topiramate (S1438), tianeptine (S5087), fluvastatin (S1909), and pitavastatin (S1759). Drugs were stored according to the manufacturer's instructions. Stock concentrations were made by dilution in DMSO. Drugs were further diluted in cell culture medium for testing in vitro.

## Tumor organoid and cell line cultures

NB tumor organoids were established from PDX in vivo models and cultured as free-floating spheres in serum-free medium with the addition of epidermal growth factor (EGF) and basic fibroblast growth factor (bFGF) as described (Persson et al, 2017; Braekeveldt et al, 2015) (Appendix Table S2). SK-N-SH (ATCC, HTB-11™) and SK-N-AS (ATCC, CRL-2137™) cell lines were cultured in RPMI 1640 (Corning, 10-040-CV) supplemented with 10% fetal bovine serum (FBS, Gibco, A5256801) and penicillin–streptomycin Solution, 100X (Corning, 30-002-CI). Models were verified by SNP analysis and regularly tested for *Mycoplasma*.

## Single-drug testing cell viability assays

LU-NB-1, LU-NB-2, and LU-NB-3 PDX-derived tumor organoids were dissociated into single cells with Accutase solution (Merck, A6964), while SK-N-SH and SK-N-AS cell lines were detached with Trypsin/EDTA (Corning, 25-053-CI). Cells were then seeded into opaque 96-well plates (Corning Inc., Corning, NY), 5000 cells per well, and treated with a range of seven concentrations (0.014–10 µM) of each of the drugs (Table 1). Cells were incubated for 3 or 7 days. Cell viability was normalized to control wells based on a metabolic readout with CellTiter-Glo (G7571; Promega, Madison, WI) luminescence. Luminescence was measured with a Synergy2 Multi-Mode plate reader (BioTek, Winooski, VT). Biological triplicates were used.

## Combination drug testing

Tumor organoids were dissociated into single cells with Accutase solution (Merck, A6964), while SKNSH and SKNAS cell lines were detached with Trypsin/EDTA (Corning, 25-053-CI). Cells were then seeded into opaque 96-well plates (5000 cells per well in a volume of 80 µl). The cells were treated with 10 µl of each statin and 10 µl of each antipsychotic at varying concentrations. Tumor organoids were incubated for 3 or 7 days and analyzed for cell viability using the CellTiter-Glo assay according to the manufacturer's instructions. For optimal organoid lysis, 2 min shaking at 350 rpm followed by 10 min incubation at <50 rpm was performed before luminescence readout. Cell viability matrices were used to calculate synergy scores and most synergistic area scores (MSA) using FIMM SynergyFinder (RRID:SCR_019318, https://synergyfinder.fimm.fi)(Ianevski et al, 2020). SynergyFinder compares the observed drug combination responses to the expected responses calculated using a synergy modeling method (here Zero Interaction Potency (ZIP) method) (Yadav et al, 2015). The predicted response is compared with experimental data, and the synergy score (δ) is calculated as the percent of response beyond expectation. Total efficacy volume was calculated using SynTox-Profiler (https://syntoxprofiler.fimm.fi). Biological duplicates were

performed for tumor organoids, whereas triplicates were performed for non-*MYCN* amplified cell lines.

## Cell death assays

NB LU-NB-1 and LU-NB-2 cells ($1 \times 10^6$ cells/T25 flask) were treated with a ~ $IC_{20}$ concentration of prochlorperazine (PCZ, 1.6 µM) and pitavastatin (PIT, 2.2 µM) or a control amount of DMSO. Cells were allowed to grow for 7 days, and photos were taken at the end of the treatment time with an Infinity 1 camera (×10) and the Infinity Analyze software (Lumenera, Ottawa, Canada). Further, treated organoids were dissociated to single cells using Accutase (#A6964, Sigma-Aldrich, St. Louis, MO) and stained with propidium iodide at a working concentration of 0.02 mg/ml (PI; P3566, Invitrogen, Waltham, MA). Flow cytometry was performed with a FACS Melody flow cytometer (BD Biosciences, Franklin Lakes, NJ), and the data were analyzed using FlowJo 10.8.1 software (RRID:SCR_008520). Biological duplicates were performed.

## In vivo study (intraperitoneal, i.p.)

LU-NB-2 PDX tumor cells ($2 \times 10^6$) were suspended 3:1 in medium/Matrigel (Cat No.354234, Corning Inc., Corning, NY) and injected subcutaneously into the flanks of female NSG mice. When tumors reached >200 mm³, each mouse was allocated randomly to a group: control, PCZ, PIT, or combination (PCZ + PIT). Drugs were formulated at 1 mg/ml in 2.5% DMSO, 2.5% Tween 80, 40% PEG, 55% PBS. Mice were treated intraperitoneally (i.p.) with daily injections six times a week according to the group assignment (Appendix Fig. S2A). The control group was administered vehicle only, PCZ and PIT were given at a dose of 5 mg/kg, and the combination group received two injections, one of each drug. Tumor volumes in mm³ were measured with calipers and calculated according to the formula $V = \pi$ (length) (width)²/6. The weight of the mice was monitored three times a week throughout the study. The study was finished on day 33 of treatment or earlier if the tumor exceeded 1500 mm³. Mice were sacrificed, and tumor pieces were divided and snap-frozen or fixed in 4% paraformaldehyde and embedded in paraffin.

RNA was extracted from snap-frozen tumor pieces in the i.p. study utilizing the AllPrep DNA/RNA Mini Kit (Qiagen, Hilden, Germany). The mRNA library preparation phase was executed employing the TruSeq Stranded mRNA Library Prep kit (20020594, Illumina, San Diego, CA). Clean up steps were automized with King Fisher FLEX (18-5400620, Thermo Fisher Scientific, Waltham, MA), and incubations and PCR were with the Eppendorf Mastercycler X50s (6311000010, Eppendorf, Hamburg, Germany). Subsequently, sequencing was carried out using the NovaSeq 6000 System (20012850, Illumina) with the NovaSeq 6000 S1 Reagent Kit, 300 cycles v1.5 (20028317), Phix Control (FC-110-3001, Illumina). Reads were aligned using STAR software, and the reference genome sequence was the Human GRCh38 primary assembly from the Ensembl database with annotation (GTF) from GENCODE v33 (RRID:SCR_002344, RRID:SCR_014966). Raw counts were normalized with DESeq2. The datasets were then uploaded onto the R2: Genomic Analysis and Visualization Platform (http://r2.amc.nl) under the designation "PDX Neuroblastoma P4-combo-in-VIVO_LUNB2_2022_117 - Aaltonen - 22 -

custom - gencode33". These procedures were performed by the Center for Translational Genomics, Lund University, and Clinical Genomics Lund, SciLifeLab. Analysis on the R2 platform investigated the top 100 most differentially expressed genes (ranked by SD, $\log_2$ transformed).

## In vivo study (intratumoral, i.t.)

LU-NB-1 PDX tumor cells ($1 \times 10^6$) were suspended in medium/Matrigel (3:1) and injected subcutaneously into the flanks of female NSG mice. When tumors reached >200 mm³, each mouse was allocated randomly to a group: control, PCZ, PIT, or combination. Drugs were formulated at 3 mg/ml and 6 mg/ml in 2.5% DMSO, 2.5% Tween 80, 40% PEG, 55% PBS. Mice were treated intratumorally (i.t.) with daily injections five times a week according to the group assignment (Fig. 2F). The control group was administered vehicle only (50 µl), whereas PCZ and PIT were given at a volume of 50 µl (c = 3 mg/ml of the respective drug). In the combination group, mice received one injection of 25 µl PCZ + 25 µl PIT (c = 6 mg/ml) mixed directly before administration. The dose of PCZ and PIT was ~5 mg/kg, and the total volume of the injection was 50 µl. Tumor volumes in mm³ were measured with calipers and were calculated according to the formula $V = \pi$ (length) (width)²/6. The weight of the mice was monitored three times a week throughout the study. The study was stopped on day 30 of treatment or earlier if the tumor exceeded 1500 mm³. Mice were sacrificed and tumor pieces were divided and snap-frozen or fixed in 4% paraformaldehyde and embedded in paraffin.

## RNA sequencing in vitro

LU-NB-2 PDX cells were seeded in T25 flasks, allowed to grow for 24 h, and treated for 48 h with the combination of PCZ + PIT (1.6 µM and 6.7 µM, $n = 6$), PCZ-only (1.6 µM, $n = 6$), PIT-only (6.7 µM, $n = 6$), or DMSO ($n = 6$). The treatment resulted in a ~15% decrease in NB cell viability in the combination sample (Appendix Fig. S3B). Cell pellets were collected, and RNA was extracted using the AllPrep DNA/RNA Mini Kit (Qiagen) and sequenced on a NovaSeq 6000 System (20,012,850, Illumina). Library preparation and mRNA sequencing were performed by the Center for Translational Genomic, Lund University.

## Quality control and bulk RNA-seq data preprocessing for in vitro and in vivo studies

Raw FASTQ files of the in vitro studies underwent quality control using the FastQC tool v0.12.1 (Andrews and Wingett, 2018) (RRID:SCR_014583) and were further filtered using the Trim Galore tool v0.6.10 (RRID:SCR_011847) with custom parameters: -q 25, –stringency of 3, and -l 50. After trimming, reads were pseudoaligned using Kallisto v0.48.0 (Bray et al, 2016) (RRID:SCR_016582) in conjunction with the GRCh38 v109 *Homo sapiens* transcriptome from the Ensembl database (Cunningham et al, 2022). The transcriptome index was generated using both the cDNA and ncRNA .fa files. MultiQC v1.14 (RRID:SCR_014982) was used to summarize the results of these steps (Ewels et al, 2016). Pseudoalignment output files were further summarized to gene level using the *tximport* R package v1.30.0 (Soneson et al, 2015) with the corresponding annotation table (v109) from the *biomaRt*

package v2.58.0 (Durinck et al, 2009) (RRID:SCR_019214). The gene-count matrix was imported into a DESeqDataSet object using DESeq2 v1.42.0 (Love et al, 2014) (RRID:SCR_000154) and assessed using the recommended DESeq2 vignette workflow, along with sample outlier detection proposed by Chen et al (Chen et al, 2020). The in vivo workflow followed a similar process, with the exception of the trimming tool. Fastp (Chen, 2023) was used for trimming with the following additional parameters: --trim_front1 3, --trim_front2 3, --trim_poly_g, --detect_adapter_for_pe, --adapter_fasta, and --length_required 50. For index generation, the Ensembl GRCh38 v111 (Martin et al, 2023) (Homo sapiens transcriptome release was used with the corresponding annotation table from biomaRt for summarization.

## Analysis of differential gene expression and gene set enrichment analysis (GSEA)

The DESeq function within DESeq2 was employed for differential gene expression (DGE) analysis, with no $\log_2$ fold-change threshold applied. The statistical analysis relied on the Wald test, and the reported $P$ values underwent adjustment using the Benjamini and Hochberg method. $\log_2$ fold change (log2FC) values were subjected to shrinkage using the apeglm method (Zhu et al, 2019). Pairwise comparisons were conducted between samples treated with drug combinations and single drugs against non-treated samples. To streamline the full gene list for subsequent volcano plots and gene set enrichment analysis (GSEA), genes with missing values in the adjusted $P$ values were excluded. The resulting differentially expressed genes were obtained for each contrast. Ranked files were generated by ordering the DGE table based on the log2FC column, serving as input for GSEA with the GSEA function from the ClusterProfiler v4.10.0 R package (Wu et al, 2021). Specific parameters were adjusted, including maxGSSize = 2000, eps = 0, and nPermSimple = 10000. GSEA utilized MSigDB Hallmark gene sets from the msigdbr v7.5.1 R package. In addition, the KEGG database (RRID:SCR_012773) was employed for GSEA using the gseKEGG function from ClusterProfiler. The rank list was converted to entrez symbols using the bitr function from the clusterProfiler package (RRID:SCR_016884). The enrichment results in Appendix Fig. S3C were manually curated to focus primarily on pathways associated with metabolism. For the in vivo data, the differential gene expression analysis followed the same procedure. However, the ranked files for GSEA were generated using $-\log10$ ($P$ value) * sign (log2FC). Additionally, GSEA for the in vivo data was performed using only the MSigDB gene sets, without employing the gseKEGG function.

## Pathway and transcription factor activities inference

For transcription factor (TF) activity inference based on expression data, we leveraged the collecTRI gene regulatory network from the R package "decoupleR" v2.8.0, which encompasses a curated collection of TFs and their target genes (Müller-Dott et al, 2023). Activity was determined through a univariate linear model using the ulm function, with the top 25 TF values reported for each contrast. It is crucial to note a slight workflow variation in generating differentially expressed genes for CollecTRI inference. Adjustments included introducing the parameter countsFromAbundance = "lengthScaledTPM" during gene-level summarization

using tximport. Furthermore, genes with fewer than 4 $\log_2$ counts and those not expressed in at least 8 samples (minimum corresponding to a condition) were excluded. Genes lacking a gene symbol were also removed. The final step involved executing the differential gene expression analysis using the limma workflow (Ritchie et al, 2015), incorporating the lmFit, contrast.fit, and eBayes functions sequentially.

## siRNA transfection and cell viability assay

Dissociated LU-NB-2 PDX cells ($0.5 \times 10^6$) were seeded into wells of a six-well plate and incubated overnight. The cells were then transfected with either siRNA control (Dharmacon) or two different siRNAs targeting *HMGCR* (siRNA #1: ID s142; 5'-UAUAAUCCCUGUAAGGUAGgt-3', siRNA #2: ID s143; 5'-GGUUCGCAGUGAUAAAGGAtt-3') (Ambion, Life Technologies). After overnight transfection, cells were resuspended and seeded into opaque 96-well plates (Corning Inc.), 5000 cells per well. The remaining cells were seeded into 6-well plates. Each siRNA condition was treated with DMSO or PCZ (1 μM) for 3 days. Cell viability, in the 96-well plates, was determined with CellTiter-Glo (G7571; Promega) luminescence assay. Luminescence was measured with a Synergy2 Multi-Mode plate reader (BioTek). Cell pellets from the six-well plate were subjected to western blot analysis, as described below. Three biological replicates were used.

## Western blotting

Cells were either treated as in the in vitro RNA-seq experiment, or with PIT (2.2 μM), PCZ (1.6 μM) and PCZ + PIT combination for 48 h. Collected cell pellets, also including those from HMGCR siRNA experiment, were lysed in RIPA buffer and supplemented with complete protease inhibitor (Roche, Basel, Switzerland) and phosSTOP (Roche). Proteins were separated using SDS–polyacrylamide gel electrophoresis gels and transferred to nitrocellulose membranes (#1704271, Bio-Rad, Hercules, CA). Antibodies used were pAktSer (1:1000 RRID:AB_331168, 193H12, Cell Signaling Technology, Danvers, MA), panAkt (1:5000 RRID:AB_1147620, 40D4, Cell Signaling Technology), MYCN (1:2000; RRID:AB_2799400, # 51705, Cell Signaling Technology), SREBP2 (1:1000; RRID:AB_779079, AB30682, Abcam), SREBP1 (1:1000; E9F4O; #95879, Cell Signaling Technology), HMGCR (1:1000; RRID:AB_2786972, #MA5-31335, Thermo Fisher Scientific), PARP (1:2000; RRID:AB_2160739, #9542, Cell Signaling Technology), GAPDH (1:5000; RRID:AB_210745, #2275-PC-100, R and D Systems) and actin (1:10,000; RRID: AB_2537667, #MA5-15739-HRP, Thermo Fisher Scientific) as a loading control. Phosphorylated and total proteins from the same biological replicate were probed on separate membranes with respective loading controls. Imaging was performed using Luminata Forte Western HRP substrate (MilliporeSigma, Burlington, MA) and Amersham Imager 600 (GE Healthcare Bio-Sciences AB, Chicago, IL). Three to five biological replicates were used.

## Immunofluorescence staining and microscope acquisition

LU-NB-2 cells were seeded on glass slides on laminin (10 μg/ml; BioLamina, Sundbyberg, LN521-05). Cells were allowed to attach and were treated with PCZ 1.6 μM + PIT 2.2 μM. DMSO-treated

cells were used as a control. After 48 h treatment, cells were fixed in 4% paraformaldehyde. For immunofluorescence staining, cells were treated with a permeabilization/blocking solution (TBS supplemented with 3% BSA, Merck, 10735078001 and 0.25% Triton™ X100 Merck, T8787) for 30 min at room temperature. Primary antibody (SREBP2, 1 μg/ml, Abcam, ab30682) and secondary antibody (Goat Anti-Rabbit IgG Alexa Fluor® 488, 1:1000, Abcam, ab150077) were incubated for 1 h at RT in antibody solution (1:1 permeabilization/blocking solution:TBS). Nuclei were counterstained with DAPI 10 mM (Invitrogen, D3571). Images were acquired with Olympus DP80 LRI microscope.

## Colocalization analysis

All samples being compared were processed in parallel, and images were acquired using the same settings. Colocalization analysis was performed with QuPath (version 0.4.3). Cells were detected with the Positive cell detection function. A total number of 382–432 DAPI+ cells were counted per condition (i.e., three biological replicates, two conditions: control and PCZ 1.6 μM + PIT 2.2 μM treated cells). Nucleus fluorescence mean intensity values for SREBP2 channel was used for further analysis. Data are presented as mean ± s.e.m. of SREBP2 nucleus fluorescence mean intensity. The Shapiro–Wilk test was used to validate the assumption of normality. Statistical significance was determined using the Kolmogorov–Smirnov test for data with non-normal distribution. Statistical analyses were performed on Prism.

## Rescue experiments

LU-NB-2 PDX-derived tumor organoids were dissociated into single cells and seeded into opaque 96-well plates (Corning Inc.), 5000 cells per well, and treated with 20 μM of PIT, 5 μM of PCZ, and PCZ + PIT (6.7 + 5 μM, respectively) alone or with respective treatments with the addition of mevalonate (100 μM and 200 μM). Cells were incubated for 3 days. Cell viability was normalized to control wells based on metabolic readout with CellTiter-Glo (G7571; Promega) luminescence. Luminescence was measured with a Synergy2 Multi-Mode plate reader (BioTek). Biological triplicates were used. Mevalonate solution was prepared fresh by dissolving mevalonolactone (M4667; Sigma-Aldrich) in 1 M HEPES buffer, and the pH was adjusted to ~7 using 1 M NaOH solution. This was further diluted in medium, and pH was controlled to be neutral before addition to the cell culture.

## Filipin staining

LU-NB-2 cells were seeded on glass slides on laminin (10 μg/ml; LN521-05, BioLamina, Sundbyberg, Sweden). Cells were allowed to attach and were treated with PIT, PCZ, and PCZ + PIT or positive control U-18666A (ab133116, Abcam, Cambridge, UK). After 48 h treatment, cells were fixed in 4% paraformaldehyde and stained with filipin (1:100) according to the protocol (ab133116, Abcam).

## Live-cell imaging

To investigate cholesterol transport during drug inhibition, we performed live cell staining on LU-NB-2 cells using LysoTracker™ Red DND-99 at 75 nM (L7528, Invitrogen) and NBD-labeled free cholesterol, 1:20 (ab269448, Abcam). Cells, grown adhering to laminin (10 μg/ml; LN521-05, BioLamina), were loaded with LysoTracker, washed, labeled with cholesterol, and treated accordingly. Cells were further imaged with a Zeiss confocal microscope (Carl Zeiss, Baden-Württemberg, Germany) from treatment to 24 h to observe the effects. Pictures for analysis were collected from three biological replicates at 24 h time point. In each picture we identified 20 ROIs and measured red fluorescence intensity/ROI (Zen 3.1, Carl Zeiss, RRID:SCR_013672) and analyzed with GraphPad.

## Lipidomics analysis

LU-NB-2 PDX cells were seeded in T25 flasks (one million per flask), allowed to grow for 24 h, and treated for 48 h with the combination of PCZ + PIT (1.6 μM and 6.7 μM, $n = 6$), half of combination of PCZ + PIT, PCZ-only (1.6 μM, $n = 6$), PIT-only (6.7 μM, $n = 6$), DMSO ($n = 6$), cisplatin (5 μM), and the combination of PCZ + PIT with cisplatin. Cells were collected as pellets until extraction.

## Extraction

Lipids were extracted using the methyl tert-butyl ether (MTBE) extraction procedure (Matyash et al, 2008; Herzog et al, 2020). A mixture of internal standards (IS) in methanol (150 μL) was added to each sample of one million cells. The following lipids were used as IS: LPC 12:0 (1 μg, Avanti Polar Lipids Inc., Alabaster, AL, USA), PC 24:0 (0.5 μg, Larodan AB, Sweden), and TG 15:0-18:1-15:0-d7 (1 μg, Avanti Polar Lipids Inc., Alabaster, AL, USA). Then, MTBE (750 μL) was added, and samples were vortexed for 5 min and sonicated for 5 min. Subsequently, 100 μL of water was added, and samples were vortexed for an additional 5 min, followed by sonication for 5 min and centrifugation for 10 min at 4 °C at 10,000 rpm in an Eppendorf centrifuge 5424 R (Eppendorf). Next, 700 μL of the MTBE phase from each sample was transferred to new glass vials, and a quality control (QC) sample was prepared by mixing aliquots from all sample extracts. All samples and QC were then evaporated to dryness using an miVac Duo Concentrator (Genevac Ltd., Ipswich, UK) connected to a membrane vacuum pump (Welch by Gardner Denver, Ilmenau, Germany). Prior to LC/MS analysis, samples were reconstituted in an isopropanol/acetonitrile mixture (9:1, v/v, 50 μL) containing PC 15:0-18:1-d7 (0.5 μg per sample, Avanti Polar Lipids Inc.).

## Untargeted LC/MS lipid analysis

Lipidomics analysis was performed on an Agilent 1290 Infinity UHPLC system (Agilent Technologies, Santa Clara, CA) coupled with a timsTOF Pro 2 (Bruker Daltonics GmbH & Co. KG, Bremen, Germany). Chromatographic separation was performed on an Acquity UPLC CSH C18 column (1.7 μm, 2.1 × 75 mm; Waters Corporation, Milford, MA) equipped with an Acquity UPLC VanGuard CSH C18 pre-column (1.7 μm, 2.1 × 5 mm; Waters Corporation) as previously described(Herzog et al, 2020). The mass spectrometer was operated in parallel accumulation serial fragmentation (PASEF) scan mode in the $m/z$ range of 50–1500 $m/z$ and $1/K_0$ range of 0.55–1.82 V·s/cm². Electrospray ionization was performed in (ESI (+)) mode with end plate offset of

500 V, capillary voltage of 4500 V, nebulizer pressure of 2.2 bar, drying gas flow of 10 L/min, and drying temperature 220 °C. During each sample acquisition, a calibration mixture was injected over 15–16 min. The calibration mixture contained Agilent Low Concentration Tuning Mix and a 10 mM sodium formate solution.

## Lipidomics data processing

Acquired data were processed in Bruker Compass MetaboScape 2022b software (v9.0.1, Bruker Daltonics GmbH & Co. KG, Bremen, Germany). Recursive feature extraction was applied to detect features if they were present in at least 3 out of 54 injected samples (including blanks and QC samples) and in at least 60% of one group corresponding to a cell treatment group. Peak intensity threshold was set to 1000 counts, with a minimum 4D peak size of 100 points. For ion deconvolution, the EIC correlation threshold was set to 0.8, and mass and mobility recalibration were performed for each sample. Peak annotation was performed based on the LipidBlast library (Tsugawa et al, 2020) and the build-in rule-based lipid annotation tool in MetaboScape. Lipid species containing >20% missing data were excluded from subsequent analysis. Missing data were imputed using a conservative approach, setting them to half the lowest detected level. Following data preprocessing, the lipid dataset underwent quality control, multivariate analysis, normalization, differential analysis, and enrichment analysis. These analyses were carried out using the lipidR workflow (Mohamed and Molendijk, 2024). Notably, probabilistic quotient normalization (PQN) was employed for data normalization. Lipids showing significant differential abundance were identified using thresholds of $P$-adjusted value ≤ 5% and $\log_2$ fold change ≥1. Enrichment analysis was performed by ranking the differentially abundant lipids based on their $\log_2$ fold-change values. For each predefined lipid set, an enrichment score (ES) was calculated using the fgsea package by assessing the overrepresentation of set members at the extremes of the ranked list. The significance of the enrichment scores was determined through permutation testing, generating a null distribution of ES values. $P$ values were adjusted for multiple testing to control the false discovery rate. The lipidomics data have been deposited to MetaboLights repository with the study identifier MTBLS13246 (Yurekten et al, 2023).

## Cell state enrichment analysis

To investigate transcriptional responses following treatment with drug combinations or single drugs, we curated a comprehensive gene set representing various cell states including differentiated adrenergic (ADR), immature mesenchymal-like (MES), sympathoblast (SYMP), Schwann cell precursor-like (SCP-like), and transitional states, drawing from established sources (Van Groningen et al, 2017; Boeva et al, 2017; Bedoya-Reina et al, 2021; Gartlgruber et al, 2021; Yuan et al, 2022; Olsen et al, 2024; Mañas et al, 2022; Thirant et al, 2023; Patel et al, 2024) (Appendix Table S4).

In addition, we incorporated gene sets associated with different cell identities in low- and high-risk NB, as proposed by Bedoya-Reina et al (Bedoya-Reina et al, 2021). Employing this expansive set of gene sets, we conducted GSEA using the rank files and the packages described in the "*Analysis of differential gene expression and gene set enrichment analysis (GSEA)*" section. It is noteworthy that, considering the limited number of genes comprising the SCP-

like gene set, we adjusted the minGSSize parameter to 2 to ensure a meaningful analysis.

## In silico analyses of public patient cohorts

For bulk analysis, the SEQC cohort of 498 NB patients was used. Using this cohort, sets of genes significantly upregulated in patients classified into high- or non-high-risk groups and directly or inversely correlated with age at diagnosis were obtained. RPMs were $\log_2$-transformed, and the difference between risk groups was calculated for each gene using FDR-corrected ANOVAs. Pearson correlations were computed between gene expression and age at diagnosis with FDR-corrected $P$ values signaling the chances of obtaining the correlation coefficient in an uncorrelated dataset. Similarly, gene sets with significantly higher expression in poor or better survival cases were retrieved with the Kaplan–Meier Scanner Pro tool in R2, using FDR-correct $P$ values obtained with log-rank tests between groups of patients with different event free-survival probabilities. Significant genes were selected with an FDR threshold of 0.01. Gene enrichment was further calculated with one-sided Fisher's exact tests corrected with the Benjamini–Hochberg approach. Using the same approach, genes significantly differentially expressed between ganglioneuroblastoma and NB were obtained for the TARGET cohort of 161 patients.

Raw microarray data files were obtained from GSE16237, GSE13136, GSE73537, and GSE3446. For the GSE3446 dataset, preprocessed normalized summarized experiments were utilized. Quality exploration of the raw unprocessed datasets was conducted using the arrayQualityMetrics package (Kauffmann et al, 2009) (RRID:SCR_001335). Samples were selected based on specific criteria for each dataset (Appendix Table S1): GSE16237: *MYCN* amplified Stage 1, 2, 3, 4, and 4S; GSE13136: 11q deletions with non-*MYCN* amplification, 1p deletions with *MYCN* amplifications, and samples with no aberrations; GSE73537: *MYCN* amplified samples; GSE3446: primary tumor samples.

Normalization of each unprocessed raw dataset was performed using the RMA algorithm. Following normalization, intensity-based filtering was applied, removing transcripts with intensities below a manual threshold of 4 in at least as many arrays as the smallest experimental group. Additionally, probes with multiple mappings were excluded from the analysis. Differential expression analysis was conducted using the limma package (RRID:SCR_010943) (Ritchie et al, 2015), with design matrices defined as follows: GSE16237: *MYCN* amplified Stage 3, 4 (High Risk) vs. *MYCN* amplified Stage 1, 2, 4S (Low Risk); GSE13136: 11q deletions with non-*MYCN* amplification (High Risk 1) vs. samples with no aberrations (No Aberrations), and 1p deletions with *MYCN* amplifications (High Risk 2) vs. samples with no aberrations (No Aberrations); GSE73537: *MYCN* amplified samples Late Stage (LS) vs. *MYCN* amplified samples early stage (ES); GSE3446: primary tumor samples that relapse (TR) vs. primary tumor samples that do not relapse (TnR), with Seeger1 representing the U133A chip and Seeger2 representing the U133B chip. GSEA was performed using a ranked file generated by ordering the DGE table based on a ranking column, calculated as the product of $-\log10$ ($P$ value) * sign (logFC). The GSEA function from the ClusterProfiler v4.10.0 R package was employed, utilizing MSigDB Hallmark gene sets from the msigdbr v7.5.1 R package.

## In vitro treatment with PCZ + PIT combination together with chemotherapy

LU-NB-1 and LU-NB-2 PDX-derived tumor organoids were dissociated into single cells and seeded on laminin in opaque 96-well plates (Corning Inc.), 5000 cells per well, and treated with DMSO, PCZ + PIT, or cisplatin for 48 h. Then, media were exchanged to another treatment and incubated for another 48 h. Cell viability was measured with CellTiter-Glo (G7571; Promega) luminescence. Luminescence was measured with a Synergy2 Multi-Mode plate reader (BioTek). At least three biological replicates were used.

## In vivo study i.t.: combination + COJEC (short-term)

LU-NB-1 PDX cells ($1 \times 10^6$) were injected subcutaneously into NSG females as above. When tumors reached 200–300 mm³, mice were randomized into treatment groups: control, PCZ + PIT combination, and PCZ + PIT combination + COJEC (Fig. 6A).

PCZ + PIT (total volume 50 µl; 5 mg/kg of each drug) was administered i.t. daily five times per week, as described for the i.t. study. The injection site was altered around the tumor to assure equal distribution across the tumor. COJEC treatment was given i.p. and consisted of five different chemotherapies distributed in cycles over one week (Fig. 6A); Day 1— cisplatin + vincristine, Day 3—etoposide + cyclophosphamide, Day 5— carboplatin (see Fig. 6A for doses). All COJEC drugs were dissolved in saline. Treatment was administered for 10 days, after which tumor pieces were divided and snap-frozen or fixed in 4% paraformaldehyde and embedded in paraffin. Mouse weights and tumor volumes were measured three times per week throughout the study. RNA was extracted from snap-frozen tumor pieces in the short-term i.p. study, and sequencing was conducted as above. These procedures were performed by the Center for Translational Genomics, Lund University, and Clinical Genomics Lund, SciLifeLab. Reads were aligned using STAR software (RRID:SCR_004463), and the reference genome sequence was from the Ensembl database, the human GRCh38 primary assembly, and the annotation (GTF) from GENCODE v33. Raw counts were normalized using DESeq2.

## Multispectral staining, acquisition, and analysis

PDX tumors were fixed in 4% paraformaldehyde (PFA) for 24–48 h post-excision. Following fixation, tissues were dehydrated using a Thermo Microm STP 120 Tissue Processor through a graded ethanol series (70% to absolute ethanol), cleared in xylene and embedded in paraffin. Tumors were sectioned to 4 µm thickness using Leica SM 2000R and baked on 62 °C for 2 h and 37 °C overnight. A multiplexed IF detection panel was designed and subsequently performed using the Akoya Biosciences protocol (Akoya Biosciences, Marlborough, USA, SKU NEL861001KT). The primary antibodies targeted TH (1:1000, Abcam, Cambridge, UK, Ab112), CD44 (1:500, Abcam, Ab157107), SOX9 (1:500, Abcam, Ab76997), NCAM (1:50, Leica Biosystems, Newcastle, UK, NCL-L-CD56-504), and PHOX2B (1:1000, Abcam, Ab183741), details described in Appendix Table S5. The multiplex antibody panel scanning was performed on PhenoImager HT 2.0 (Akoya Biosciences) at ×20. A spectral fluorophore was used for each primary antibody and a synthetic library combined with the extraction of tissue-specific autofluorescence spectrum was prepared using the InForm software (version 3.0, Akoya Biosciences) to enable optimal multispectral unmixing All samples being compared were processed in parallel and images were acquired using the same settings. Fluorescence classification and colocalization were performed using QuPath (version 0.5.0). For each biological replicate, three regions of interest (ROIs) with the highest NCAM expression were selected (the only exception being C1, with two ROIs qualitatively approved for analysis). Two biological replicates (tumors) were analyzed per condition. Cells were counted with the Cell detection function. A total number of 2000–7000 DAPI+ cells were counted per ROI and classified within each ROI using intensity-based classifiers for the markers TH, SOX9, NCAM, and PHOX2B. CD44 was not detected in NCAM-positive cells and was therefore excluded from further analysis.

Data are presented as the mean of marker + /NCAM+ cells ± s.d. The Kolmogorov–Smirnov test was used to validate the assumption of normality. Statistical significance was determined using two-way ANOVA and Bonferroni's nonparametric test for data with non-normal distribution.

## Survival study: PCZ + PIT combination + COJEC

Injection of LU-NB-1 PDX cells ($1 \times 10^6$) was performed subcutaneously in NSG females as above. When the tumor reached 200–300 mm³, the mice were randomized into treatment groups: control, PCZ + PIT combination, COJEC, or PCZ + PIT combination + COJEC (Fig. 6G).

PCZ + PIT (total volume 50 µl; 5 mg/kg of each drug) was administered i.t. daily five times per week, as described for the i.t. study (Fig. 2F). The injection site was altered around the tumor to assure equal distribution across the tumor (Fig. EV2E). All mice received i.t. injections on treatment days, i.e., vehicle (2.5% DMSO, 2.5% Tween 80, 40% PEG, 55% PBS) was injected in tumors in the control and COJEC-only groups. COJEC treatment was given i.p. and consisted of five different chemotherapies distributed in cycles over 1 week (Fig. 6G); Day 1—cisplatin + vincristine, Day 3—etoposide + cyclophosphamide, Day 5—carboplatin (see Fig. 6G for doses). All COJEC drugs were dissolved in saline. Treatment breaks in the COJEC protocol were added if the mouse weight dropped below 90%.

Treatment was administered for 30 days, after which tumors were allowed to regrow. Mouse weights and tumor volumes were measured three times per week throughout the study. Mice were sacrificed when the tumors reached 1500 mm³. Tumors tissue was collected and snap-frozen or fixed in 4% paraformaldehyde and embedded in paraffin.

## Statistical analysis

Data were analyzed in GraphPad Prism v9.3.0 for Windows (RRID:SCR_002798, GraphPad Software, San Diego, CA) or Excel 2016 (Microsoft, RRID:SCR_016137). In vivo groups were compared using one-way ANOVA on the last day of all mice being alive followed by Tukey's multiple comparison test, unless stated otherwise. Survival analysis was performed with Kaplan–Meier analysis and log-rank tests between indicated groups. In cell state enrichment analysis, groups were compared with the Welch $t$ test. Protein expression in WB analysis between

**The paper explained**

**Problem**

Over half of children with high-risk neuroblastoma (NB) exhibit either initial or acquired treatment resistance to current clinical therapies. Thus, it is urgent to find novel treatment strategies against resistant NB. However, conventional drug discovery is slow, expensive, often fails in practice, and consequently falls short in addressing pediatric and rare conditions.

**Results**

Here, we applied in silico prediction methods including machine-learning algorithms to identify clinically approved drugs for repurposing against NB. In vitro drug testing revealed a synergistic drug combination of two approved drugs (statins and prochlorperazine) which led to NB cell death and altered cholesterol metabolism. In vivo, the novel drug combination together with chemotherapy extended survival and outperformed chemotherapy alone.

**Impact**

This study shows the potential of integrating machine learning and drug repurposing with patient-derived tumor models to develop improved treatments for high-risk NB. Therapeutic targeting of cholesterol metabolism can sensitize NB to standard-of-care chemotherapy. Importantly, both statins and prochlorperazine are already approved for clinical use, which could lead to shorter lead time from preclinical validation to clinical application.

treatment groups was compared using one-way ANOVA followed by Tukey's multiple comparison test. Colocalization analysis of SREBP2 fluorescence signal in the nucleus was performed with Kolmogorov–Smirnov test for data with non-normal distribution. Mevalonate rescue experiments were analyzed using one-way ANOVA test and treatment groups were compared to control with Dunnet's multiple comparison test. Live cell imaging experiment results were analyzed with one-way ANOVA followed by Tukey's multiple comparison test between all groups.

For lipidomics analysis, a permutation test was used to calculate $P$ values, corrected with Benjamini–Hochberg test. In vitro viability cisplatin and/or combination treatment was compared between groups using one-way ANOVA with Tukey's multiple comparisons test. Analysis of marker expression in NCAM+ cells after treatment was performed with the Kolmogorov–Smirnov to validate the assumption of normality. Statistical significance was thereafter determined using two-way ANOVA and Bonferroni's nonparametric test for data with non-normal distribution.

### Graphics

Figure 1 and synopsis were created with BioRender.com.

## Data availability

The datasets produced in this study are available in the following databases: -Bulk RNAseq: BioStudies: RNA-seq of tumor organoids from patient-derived xenograft (PDX, LU-NB-2) of high-risk, MYCN-amplified neuroblastoma treated with prochlorperazine (PCZ), pitavastatin (PIT), the combination of both drugs, and untreated controls. E-MTAB-14846. Bulk RNAseq: BioStudies:

RNA-seq of subcutaneous patient-derived neuroblastoma xenografts (LU-NB-1) - Comparison of Prochlorperazine + Pitavastatin (PCZ + PIT) combination alone to one combined with standard-of-care COJEC protocol (PCZ + PIT + COJEC). E-MTAB-14902. Multispectral images: BioImage Archive: Repurposing statins and phenothiazines to treat chemoresistant neuroblastoma. S-BIAD2273 (https://www.ebi.ac.uk/biostudies/bioimages/studies/S-BIAD2273?key=dc704aef-22f7-4173-97eb-f04c38389400). The lipidomics data have been deposited to MetaboLights repository with the study identifier MTBLS13246.

The source data of this paper are collected in the following database record: biostudies:S-SCDT-10_1038-S44321-025-00349-6.

## Peer review information

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

## Acknowledgements

This work is dedicated to Neil T Thompson, a great scientist and colleague who passed away during the finalization of this work. We thank the Center for Translational Genomics (CTG), Lund University, and Clinical Genomics Lund, SciLifeLab for providing sequencing services. Part of the computations and data handling were enabled by resources provided by the National Academic Infrastructure for Supercomputing in Sweden (NAISS), partially funded by the Swedish Research Council through grant agreement no. 2022-06725. We thank Sebastian Braun from the Division of Translational Cancer Research, Lund University for sharing his confocal microscopy expertise. Funding received for this project: ENEA- European Neuroblastoma Association (DB), aPODD Foundation (DB), Healx (DB), Swedish Cancer Society 20 0897 PjF and 23 2754 Pj (DB), Swedish Childhood Cancer Foundation grants PR2020-0018 and PR2023-0006 (DB), Swedish Research Council grants 2021-02597 and 2023-02402 (DB), Crafoord Foundation grant 20210593 (DB), Region Skåne and Skåne University Hospital Funding grant (DB), Swedish Childhood Cancer Foundation grants TJ2021-0137 and PR2021-0129 (OCBR), KI Forskningsbidrag grant 2022-01925 (OCBR), Kungliga Fysiografiska Sällskapet i Lund 41984 and 42964 (KR), Swedish Childhood Cancer Foundation grant TJ2021-0068 (JTS).

## Author contributions

**Katarzyna Radke**: Formal analysis; Investigation; Visualization; Methodology; Writing—original draft; Writing—review and editing; Designed the study. Performed in vitro and in vivo experiments Analyzed and visualized the data. Wrote and revised the manuscript. **Kristina Aaltonen**: Formal analysis; Supervision; Investigation; Visualization; Methodology; Writing—original draft; Writing—review and editing; Designed the study. Performed in vitro experiments. Analyzed and visualized the data. Wrote and revised manuscript. Supervised the study. **Erick A Muciño-Olmos**: Data curation; Formal analysis; Investigation; Visualization; Bioinformatic analysis. **Javanshir Esfandyari**: Investigation; Performed in vivo experiments. **Aleksandra Adamska**: Investigation; Performed in vitro and in vivo experiments. **Joachim T Siaw**: Formal analysis; Investigation; Visualization; Performed in vitro experiments. Analyzed and visualized the data. **Dora Adamic**: Formal analysis; Investigation; Visualization; Performed in vitro experiments. Analyzed and visualized the data. **Chiara Lago**: Formal analysis; Investigation; Visualization; Performed in vitro experiments. Analyzed and visualized the data. **Adriana Mañas**: Investigation; Performed in vivo experiments. **Alexandra Seger**: Investigation; Performed in vitro experiments. **Karin Hansson**: Investigation; Performed in vitro experiments. **Oksana Rogova**: Formal analysis; Investigation; Performed lipidomic experiments and analysis. **Sophie Lehn**: Formal analysis; Validation; Analyzed and visualized the data. **Daniel J Mason**: Data curation; Software; Formal analysis; Performed original predictions and drug selection. Performed bioinformatic analysis. **Daniel J O'Donovan**: Software; Formal analysis; Performed the original predictions and drug selection. **Ian Roberts**: Software; Formal analysis; Performed original predictions and drug selection. **Antonia Lock**: Software; Formal analysis; Performed original predictions and drug selection. **Jane Brennan**: Software; Formal analysis; Performed original predictions and drug selection. **Kristian Pietras**: Formal analysis; Visualization; Analyzed and visualized data. **Emma J Davies**: Software; Formal analysis; Performed original predictions and drug selection. **Peter Spégel**: Formal analysis; Investigation; Performed lipidomic experiments and analysis. **Oscar C Bedoya-Reina**: Data curation; Formal analysis; Performed bioinformatic analysis. **David Brown**: Conceptualization; Supervision; Conceptualized the study. Designed the study. Performed original predictions and drug selection. Supervised the study. **Neil T Thompson**: Software; Supervision; Performed original predictions and drug selection. **Cesare Spadoni**: Conceptualization; Supervision; Conceptualized the study. Supervised the study. **Daniel Bexell**: Conceptualization; Supervision; Methodology; Conceptualized the study. Designed the study. Supervised the study.

Source data underlying figure panels in this paper may have individual authorship assigned. Where available, figure panel/source data authorship is listed in the following database record: biostudies:S-SCDT-10_1038-S44321-025-00349-6.

## Funding

## Disclosure and competing interests statement

DBe has received research funding from Healx, ENEA, and aPODD Foundation for this project. CS is a shareholder of Oncoheroes Biosciences Inc. DM, DO, IR, AL, JB, ED, DBr, and NT are (or have been) employed at Healx. The remaining authors declare no competing interests.

# Expanded View Figures

**Figure EV1.  Single drug efficacy and PCZ + PIT combination synergy.**

(A) Dose–response curves of 3 or 7 days ($n = 2$). (B) Synergy matrix of the PCZ + PIT combination for LU-NB-1 and LU-NB-2 over 3 or 7 days ($n = 2$). (C) Response of non-*MYCN* amplified NB cell lines, SK-N-SH and SK-N-AS to Prochlorperazine and Pitavastatin over 3 or 7 days (data represent mean ± SD; $n = 3$). (D) Synergy matrix of the PCZ + PIT combination for SK-N-SH and SK-N-AS over 3 or 7 days ($n = 3$). (E) Subcutaneous LU-NB-1 PDXs tumor starting volume (control $n = 5$, PCZ $n = 4$, PIT $n = 5$, and combination $n = 6$, one-way ANOVA followed by Tukey's multiple comparisons test, n.s., boxes represent the interquartile range and whiskers indicate minimum and maximum values). (F) Average weight ratio per group. (G) Tumor size change from baseline for each mouse at day 8 (last day of all mice alive). (H) Tumor growth of individual mice over time. PCZ prochlorperazine, TFP trifluoperazine, PIT pitavastatin, i.t. intratumoral injection. Source data are available online for this figure.

◀

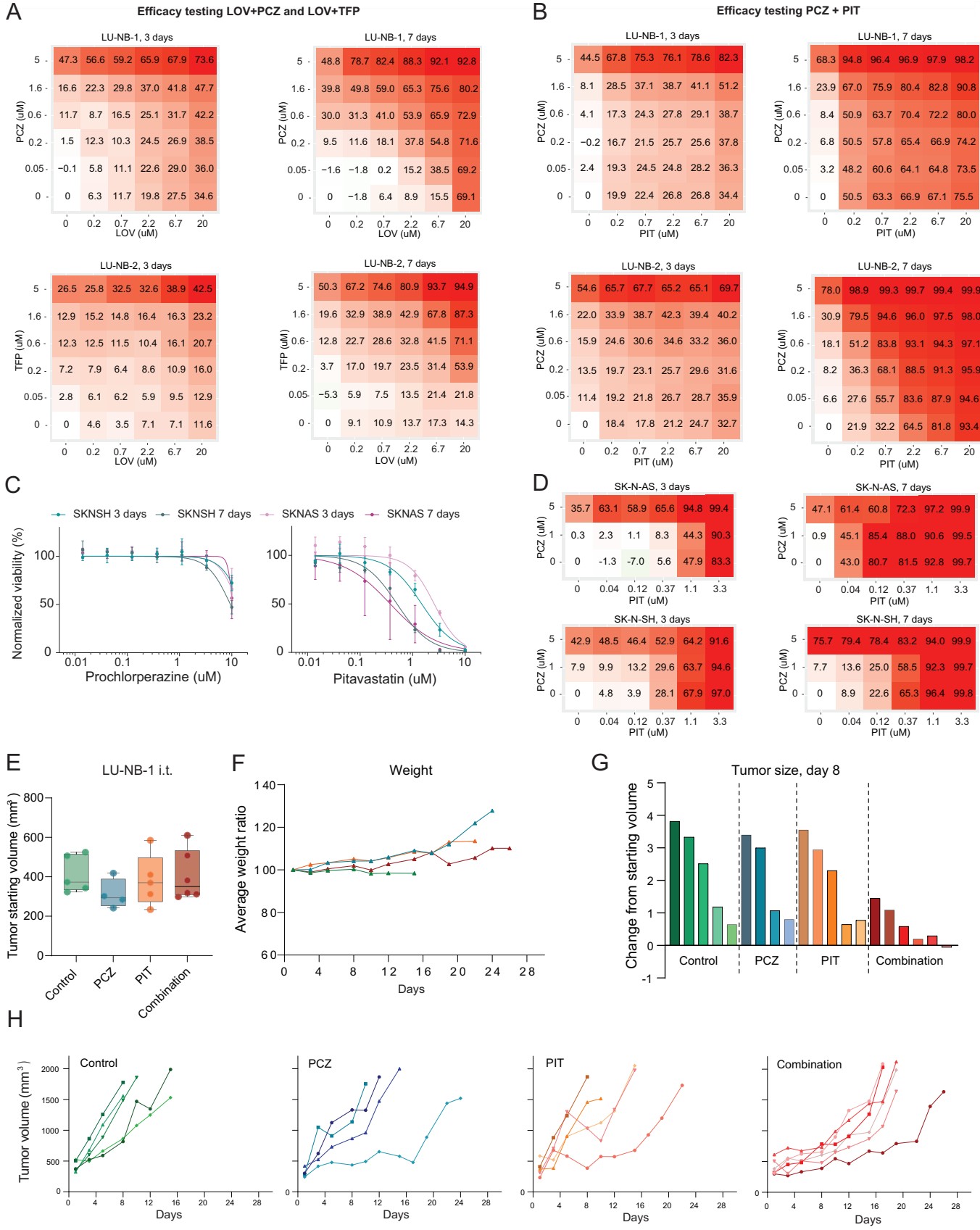

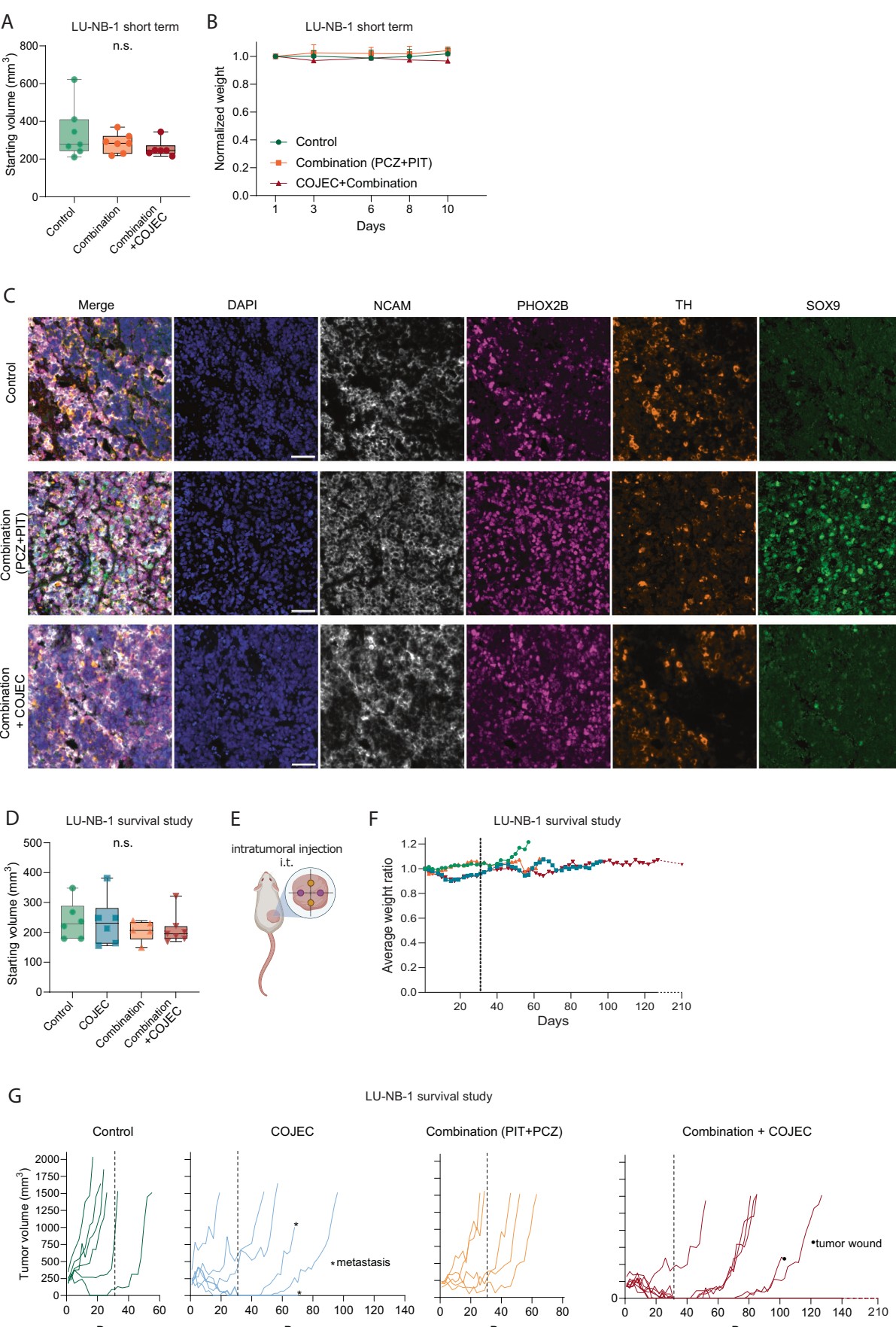

◀ **Figure EV2. PCZ + PIT enhances standard-of-care COJEC compared to COJEC alone.**

Short-term study. (**A**) LU-NB-1 PDXs tumor starting volume (control $n = 7$, combination $n = 7$, and combination + COJEC $n = 6$, one-way ANOVA followed by Tukey's multiple comparisons test, n.s. boxes represent the interquartile range and whiskers indicate minimum and maximum values). (**B**) Average weight ratio per group (control $n = 7$, combination $n = 7$, and combination + COJEC $n = 6$; data represent mean ± SD). (**C**) Representative images of single marker expression and of co-expression for each treatment group. NCAM = grey, PHOX2B = magenta, TH = orange, SOX9 = green. Scale bar = 50 μm. Survival study. (**D**) LU-NB-1 PDXs tumor starting volume (control $n = 6$, COJEC $n = 6$, combination $n = 5$, comb + COJEC $n = 7$, one-way ANOVA followed by Tukey's multiple comparisons test, n.s. boxes represent the interquartile range and whiskers indicate minimum and maximum values). (**E**) Schematic representation of intratumoral injections. (**F**) Average weight ratio per group throughout the study. Treatment stop marked with dotted line. (**G**) Tumor growth of individual mice over time for each of the groups. Treatment stop marked with dotted line. Combination=PCZ + PIT; COJEC- see Fig. 6G; PCZ prochlorperazine, PIT pitavastatin, i.t. intratumoral. Source data are available online for this figure.

