## [Peer Review File · EMBO Molecular Medicine]

Repurposing statins and phenothiazines to treat chemoresistant neuroblastoma

Katarzyna Radke, Kristina Aaltonen, Erick Muciño-Olmos, Javanshir Esfandyari, Aleksandra Adamska, Joachim Siaw, Dora Adamic, Chiara Lago, Adriana Mañas, Alexandra Seger, Karin Hansson, Oksana Rogova, Sophie Lehn, Daniel Mason, Daniel O'Donovan, Ian Roberts, Antonina Lock, Jane Brennan, Kristian Pietras, Emma Davies, Peter Spéjel, Oscar Bedoya Reina, David Brown, Neil Thompson, Cesare Spadoni, and Daniel Bexell

Corresponding author: Daniel Bexell (daniel.bexell@med.lu.se)

Review Timeline:

Submission Date:	4th Dec 24
Editorial Decision:	17th Jan 25
Revision Received:	9th Sep 25
Editorial Decision:	24th Sep 25
Revision Received:	4th Nov 25
Accepted:	14th Nov 25

Editor: Lise Roth

Transaction Report:

17th Jan 2025

Dear Dr. Bexell,

Thank you for the submission of your manuscript to EMBO Molecular Medicine, and please accept my apologies for the delay in getting back to you as we were waiting for one referee report. Unfortunately, referee #3 has not yet gotten back to us despite several chasers, but given that both referees #1 and #2 provide similar recommendations, we prefer to make a decision now in order to avoid further delay in the process. Should referee #3 provide a report, we will send it to you, with the understanding that we will not ask you for extensive experiments in addition to the ones required in the enclosed reports from referees #1 and #2.

As you will see from the reports below, the referees acknowledge the interest of the study and are overall supporting publication of your work pending appropriate revisions. Upon further cross-commenting with the referees, we agreed that revisions could focus on in vitro experiments, and that in vivo testing of combination treatment (PCZ+PIT followed by COJEC treatment) would NOT be required for further consideration of the manuscript.

Addressing the reviewers' concerns in full will be necessary for further considering the manuscript in our journal, and acceptance of the manuscript will entail a second round of review. EMBO Molecular Medicine encourages a single round of revision only and therefore, acceptance or rejection of the manuscript will depend on the completeness of your responses included in the next, final version of the manuscript. For this reason, and to save you from any frustrations in the end, I would strongly advise against returning an incomplete revision.

We are expecting your revised manuscript within three to four months, if you anticipate any delay, please contact us.

We require:

4) A .docx formatted letter INCLUDING the reviewers' reports and your detailed point-by-point responses to their comments. As part of the EMBO Press transparent editorial process, the point-by-point response is part of the Review Process File (RPF), which will be published alongside your paper.

5) A complete author checklist, which you can download from our author guidelines (<https://www.embopress.org/page/journal/17574684/authorguide#submissionofrevisions>). Please insert information in the checklist that is also reflected in the manuscript. The completed author checklist will also be part of the RPF.

6) All Materials and Methods need to be described in the main text using our 'Structured Methods' format. According to this format, the Methods section includes a Reagents and Tools Table (listing key reagents, experimental models, software and relevant equipment and including their sources and relevant identifiers) followed by a Methods and Protocols section describing the methods, ideally using a step-by-step protocol format. The aim is to facilitate adoption of the methodologies across labs. Please download and fill our Reagents and Tools Table template (.docx), which you can find in our author guidelines: <https://www.embopress.org/page/journal/14693178/authorguide#structuredmethods>.

<https://www.embopress.org/doi/10.15252/msb.20178071>

7) Please note that all corresponding authors are required to supply an ORCID ID for their name upon submission of a revised manuscript.

8) It is mandatory to include a 'Data Availability' section after the Materials and Methods. Before submitting your revision, primary datasets produced in this study need to be deposited in an appropriate public database, and the accession numbers and database listed under 'Data Availability'. Please remember to provide a reviewer password if the datasets are not yet public (see <https://www.embopress.org/page/journal/17574684/authorguide#dataavailability>).

9) For data quantification: please specify the name of the statistical test used to generate error bars and P values, the number (n) of independent experiments (specify technical or biological replicates) underlying each data point and the test used to calculate p-values in each figure legend. The figure legends should contain a basic description of n, P and the test applied. Graphs must include a description of the bars and the error bars (s.d., s.e.m.). Please provide exact p values.

10) Our journal encourages inclusion of *data citations in the reference list* to directly cite datasets that were re-used and obtained from public databases. Data citations in the article text are distinct from normal bibliographical citations and should directly link to the database records from which the data can be accessed. In the main text, data citations are formatted as follows: "Data ref: Smith et al, 2001" or "Data ref: NCBI Sequence Read Archive PRJNA342805, 2017". In the Reference list, data citations must be labeled with "[DATASET]". A data reference must provide the database name, accession number/identifiers and a resolvable link to the landing page from which the data can be accessed at the end of the reference. Further instructions are available at .

11) We replaced Supplementary Information with Expanded View (EV) Figures and Tables that are collapsible/expandable online. A maximum of 5 EV Figures can be typeset. EV Figures should be cited as 'Figure EV1, Figure EV2' etc... in the text and their respective legends should be included in the main text after the legends of regular figures.

12) The paper explained: EMBO Molecular Medicine articles are accompanied by a summary of the articles to emphasize the major findings in the paper and their medical implications for the non-specialist reader. Please provide a draft summary of your article highlighting

- the medical issue you are addressing,

- the results obtained and

- their clinical impact.

13) Author contributions: CRedit has replaced the traditional author contributions section because it offers a systematic machine readable author contributions format that allows for more effective research assessment. Please remove the Authors Contributions from the manuscript and use the free text boxes beneath each contributing author's name in our system to add specific details on the author's contribution. More information is available in our guide to authors.

Please also suggest a visual abstract to illustrate your article as a PNG file 550 px wide x 300-600 px high. A cropped portion of this image will serve as thumbnail for the table of content on our webpage.

16) As part of the EMBO Publications transparent editorial process initiative (see our Editorial at <http://embomolmed.embopress.org/content/2/9/329>), EMBO Molecular Medicine will publish online a Review Process File (RPF)

to accompany accepted manuscripts.

In the event of acceptance, this file will be published in conjunction with your paper and will include the anonymous referee reports, your point-by-point response and all pertinent correspondence relating to the manuscript. Let us know whether you agree with the publication of the RPF and as here, if you want to remove or not any figures from it prior to publication. Please note that the Authors checklist will be published at the end of the RPF.

I look forward to receiving your revised manuscript.

Yours sincerely,

Lise Roth

***** Reviewer's comments *****

Referee #1 (Remarks for Author):

This manuscript demonstrates that the combination of statins and phenothiazines, based on the idea of drug repurposing, may be a new treatment option for refractory neuroblastoma. In a mouse model injected with PDOs derived from refractory neuroblastoma, the combination of these drugs and COJEC showed significant tumor suppression and prolonged survival without causing weight loss. By combining the transcriptome and lipidome, the authors found that these drugs work by 1) cholesterol deprivation and 2) differentiation from an undifferentiated MES-like cell state to an adrenergic cell state. This manuscript contains important findings that could potentially revolutionize the standard of care in refractory neuroblastoma. However, the potential appeal of this manuscript is undermined by inadequate analysis of molecular mechanisms, patient selection, and dosing regimen.

Major comments

- 1) In Figure 3, the authors use the transcriptome to identify the intracellular pathways that are the action sites of PCZ+PIT and identify MYC/MYCN and SREBF1/2 as the most affected transcription factor candidates. However, actual changes in protein expression levels, subcellular localization, and transcriptional activity of these transcription factors have not been examined. MYCN is a marker gene that defines the transitional subtype used in Figures 4B and 4C (Yuan et al., Cell Rep 2022), and MYCN regulates the MYCN-amplified subtype of the four subtypes of neuroblastoma, which are defined by super enhancers (Gartlgruver, Nature Cancer, 2021). Furthermore, decreased MYCN expression is also an indicator of favorable response to COJEC (Manas et al., Science Advances 2022). Given that MYCN regulates cholesterol metabolism in neuroblastoma, the two points of action of PCZ+PIT that the authors claim (cholesterol deficiency and cell state changes) could be explained by a common molecular mechanism, MYCN/MYC inhibition, but this has not been tested. The same is true for SREBF1/2. The molecular mechanism shown in Figure 3D is an excessive speculation unless the expression levels, subcellular localization, and transcriptional activities of these transcription factors are examined.
- 2) In Figure 5C and 5D, differentiation from MES to ADRN is estimated using transcriptome. However, in Figure 5D, they did not detect the changes expected from the in vitro results in Figure 4. Although the authors discuss this straightforwardly in the text (lines 342, 343), there is no detailed discussion of this result. This result should be experimentally examined by single-cell analysis or immunostaining. In immunostaining, markers such as PHOX2B, SOX9(MES), LGR5(MES), TH(ADR), and MYCN have been reported (Manas et al., Science Advances 2022). The combination of these markers with immune cell markers may enable a detailed description of the molecular mechanisms described above.
- 3) Related to 2), increased clonal diversity is a major molecular mechanism of chemotherapy resistance in COJEC (Manas et al., Science Advances 2022), and whether PCZ+PIT can suppress clonal diversity should be examined.

4) Although the authors used only MYCN-amplified neuroblastoma in their experimental data, they treated both MYCN-amplified and non-amplified neuroblastoma as high-risk neuroblastoma in their in silico data analysis. They also discuss the possibility that the combination of these drugs may be effective in MYCN non-amplified neuroblastoma (lines 459-461), but there is no experimental data to support this. It remains to be clarified whether the combination of these agents is effective only in MYCN amplified NBs or also in high-risk MYCN non-amplified NBs. At least, they should test whether in vitro drug sensitivity differs between MYCN amplified NB and non-amplified NB cell lines. It should also be examined whether MYC expression levels, transcriptional activity, cholesterol deficiency and ADRN-like differentiation are detected in MYCN non-amplified NB- cell lines sensitive to PCZ+PIT. In the data analysis, Fig 4D and Fig S5A should be reanalyzed separately for each MYCN status.

5) As the authors recognize, the greatest limitation is that the method of administration is only effective in the context of intratumoral treatment; drug repurposing is beneficial because it allows the use of drugs with known side effects and pharmacokinetics, but the results suggest that pharmacokinetics need to be improved. Thus, there is a need to synthesize new compounds with enhanced tumor accumulation using these drugs as seed compounds, and to reevaluate the safety and side effects of these new compounds. In other words, the usefulness of drug repurposing discussed in the introduction and discussion of this manuscript is likely to be compromised. The authors do not discuss methods to increase tumor accumulation without changing the chemical structure of PCZ or PIT, although such methods exist. In addition, these drugs are already in clinical use for the treatment of other diseases, and it has not been discussed whether the maximum dose and route of administration of these drugs in humans is similar to the dose and route of administration in the mouse model. Furthermore, in vitro, prior treatment with the PCZ+PIT combination showed enhanced cytotoxic effect (Fig. 4E), while in vivo, COJEC and PCZ+PIT were administered simultaneously (Fig. 5E-H). Considering that drug-drug interactions increase the risk of unexpected side effects, the combination of five drugs and PCZ+PIT is likely to increase the risk of side effects other than weight loss. The efficacy of the PCZ+PIT followed by COJEC treatment should be tested in vivo.

Minor comments

- 1) Without reading the main text, it is impossible to understand the meaning of "dual-hit mechanism" and "phenotype switch" described in the abstract. These should be changed to more specific descriptions.
- 2) In Fig. 1 legend, too, "dual-hit mechanism" is not immediately understandable to readers. In particular, the "dual-hit hypothesis" is used to explain completely different etiologies in the fields of Parkinson's disease, schizophrenia, and cardiovascular disease, and is likely to cause confusion. The "dual-hit" hypothesis in these fields defines a first hit and a second hit, which is very different from the way the term "dual-hit" is used in this manuscript. In particular, since LDL-C is associated with the first hit in cardiovascular disease, the reader may misunderstand the meaning of cholesterol control in this manuscript as being related to LDL-C at the beginning of their reading. It is appropriate to simply state, such as cholesterol deficiency.
- 3) The details of the drug selection process in lines 141-148 should be provided. How many drugs were initially identified as candidates, how many of each were eliminated, and for what reasons, and finally the 12 compounds in Table 1 were selected.
- 4) Table S3 IC50 concentration units should be described.
- 5) The drug concentration in Figure 2D should be described.
- 6) The meaning of QD in Figure 2F should be described.
- 7) The calculation method of total efficacy volume in Figure 2B should be described.
- 8) In Figure 2H, there are unclear characters under "days".
- 9) In Line 298, "undifferentiated MES-like signatures" may be a new term defined by the authors to equate the transitional subtype of Yuan et al. with the MES subtype of Manas et al. . However, Yuan et al. distinguish transitional subtype from MES subtype (Yuan et al., Cell Rep 2022). The new term should be clearly stated and defined.
- 10) It is not clear to the reader what the SYMP-Patel signature in Line 298 means.
- 11) The possible relationship between the dual-hit mechanism and the phenotype switch should be discussed.

Referee #2 (Remarks for Author):

The manuscript "Repurposing statins and phenothiazines to treat chemoresistant neuroblastoma" by Radke et al explores the potential to use statins and phenothiazines in the treatment of MYCN-amplified neuroblastoma. The authors applied in silico prediction to identify approved drugs for repurposing against neuroblastoma which identified statins and phenothiazines. The work validates these drugs as single treatments and in combination, as well as together with clinically relevant chemotherapy protocol, in MYCN-amplified tumor models in vitro and in vivo. The manuscript is well written, the data is solid and convincing, and clearly presented, and the discussion is relevant and appropriate. The work has strong translational value, investigating drugs already used in the clinic. The findings would be of great interest to the neuroblastoma and cancer community.

I do have some recommendations, comments, and questions for the authors, outlined below.

1. The study uses neuroblastoma organoid models in vitro, the authors describe that the cells are cultured as free-floating spheres in serum-free medium with the addition of EGF and bFGF, is anything known in regard to the cells' clonal heterogeneity? To differentiate them from cell line models. How do the cells grow in a plate, do they attach or do they form spheroids? How was the viability determined after 7 days if the cells grow in spheroids/clusters? Since CellTiter Glo is described to poorly penetrate 3D spheroids.

2. siRNA experiments downregulating HMGCR, in combination with phenothiazines (e.g., PCZ) would strengthen the study in terms of further confirming the mechanism of action for the synergy.
3. As the authors point of themselves, the drugs are only tested in MYCN-amplified models in vitro and in vivo, please make this clearer in the conclusions of the paper, for example in Fig 6.
4. Please specify what is shown in each figure, i.e., mean, measurement of variation, number of experiments, etc. Also, please clarify what is shown in Figure 2B to the left, define MSA synergy score and efficacy score in the figure legend (MSA is explained in the text but not efficacy score).
5. How was synergy defined? In SynergyFinder it is suggested with >10 . The authors claim that the combinations induce synergy in Fig. 2B-C and S2B, which is mostly true for 7 days, but not 3 days according to the limit suggested by SynergyFinder, more correctly would be to state additive or synergetic effects, or specify that the synergetic effects are more pronounced at the longer time point. Furthermore, do the authors have any possible explanation for the improved synergetic effects for longer incubation time?

Minor comments

6. The authors propose that the lack of in vivo activity is due to first pass metabolism. Intraperitoneal injection (ip) bypasses first-pass metabolism from the gut, but is still subject to first-pass metabolism by the liver, so it is a plausible explanation, however if so, only owing to metabolism in the liver.
7. I believe that the authors might have done a mistake with the formula for calculating in vivo tumor volume, given at row 620 and 650 and written: $V = \pi(\text{length})(\text{width})^2$.

***** Reviewer's comments *****

Referee #1 (Remarks for Author):

This manuscript demonstrates that the combination of statins and phenothiazines, based on the idea of drug repurposing, maybe a new treatment option for refractory neuroblastoma. In a mouse model injected with PDOs derived from refractory neuroblastoma, the combination of these drugs and COJEC showed significant tumor suppression and prolonged survival without causing weight loss. By combining the transcriptome and lipidome, the authors found that these drugs work by 1) cholesterol deprivation and 2) differentiation from an undifferentiated MES-like cell state to an adrenergic cell state. This manuscript contains important findings that could potentially revolutionize the standard of care in refractory neuroblastoma. However, the potential appeal of this manuscript is undermined by inadequate analysis of molecular mechanisms, patient selection, and dosing regimen.

Major comments

Comment 1

In Figure 3, the authors use the transcriptome to identify the intracellular pathways that are the action sites of PCZ+PIT and identify MYC/MYCN and SREBF1/2 as the most affected transcription factor candidates. However, actual changes in protein expression levels, subcellular localization, and transcriptional activity of these transcription factors have not been examined. MYCN is a marker gene that defines the transitional subtype used in Figures 4B and 4C (Yuan et al., Cell Rep 2022), and MYCN regulates the MYCN-amplified subtype of the four subtypes of neuroblastoma, which are defined by super-enhancers (Gartlgruver, Nature Cancer, 2021). Furthermore, decreased MYCN expression is also an indicator of favorable response to COJEC (Manas et al., Science Advances 2022). Given that MYCN regulates cholesterol metabolism in neuroblastoma, the two points of action of PCZ+PIT that the authors claim (cholesterol deficiency and cell state changes) could be explained by a common molecular mechanism, MYCN/MYC inhibition, but this has not been tested. The same is true for SREBF1/2. The molecular mechanism shown in Figure 3D is an excessive speculation unless the expression levels, subcellular localization, and transcriptional activities of these transcription factors are examined.

We appreciate the reviewer's insightful comment regarding the need for a more detailed analysis of the proposed molecular mechanisms. In response, we have expanded our investigation and included additional functional experiments, which are now presented in the revised manuscript as **Figure 3** and the newly added **Figure 4**. A summary of these new experiments is provided below.

First, *MYCN* is an established driver gene in *MYCN*-amplified neuroblastoma. Given that our study primarily involves *MYCN*-amplified models, *MYCN* expression is not an appropriate marker for identifying a transitional subtype. In contrast, the study by Yuan et al. (2022), which included predominantly non-*MYCN*-amplified tumors (4 out of 5), reported *MYCN* upregulation in the transitional subtype, an observation largely driven by a single ganglioneuroblastoma sample with high *MYCN* expression and a transitional cell state.

In our data, the observed downregulation of *MYCN* (**Fig. 3B**) is more likely associated with reduced proliferation and an overall favorable treatment response, consistent with findings from Manas et al. (2022). This is further supported by our new WB results which show stable *MYCN* protein levels during treatment (**Fig. 3D,E; 4B; S3G**). In contrast, changes in proteins involved in cholesterol metabolism were more pronounced (see below).

Moreover, drug treatment responses in non-*MYCN* amplified cell lines were comparable to those observed in the *MYCN*-amplified *in vitro* models (Fig. EV1C, D). Collectively, our data suggest that *MYCN* activity is not the primary driver of PCZ+PIT treatment response. Instead, reduced *MYCN* gene expression is a downstream consequence, associated with decreased proliferation and increased cell death. We have now expanded the description of these results in the text related to Figures 3 and new 4.

We agree with the reviewer that additional experiments were needed to show activation of SREBP1/2 as well as downstream effects. In response, we have now investigated the mechanism of action and regulation of observed processes under PCZ+PIT treatment.

Firstly, we used CollecTRI to infer gene regulatory networks from our transcriptomic data. CollecTRI is a comprehensive resource for gene regulatory network analysis that enables inference of transcription factor (TF) activities by integrating diverse datasets, thereby improving coverage and accuracy in estimating TF activities. Our CollecTri analysis revealed upregulation of *SREBF1* and 2 activity upon drug PCZ+PIT treatment (now Fig 3B and S3F). We also identified several downstream effectors and activators of *SREBF1* and 2, like *ACSL1*, *ACSS2*, *INSIG1*, and *FDPS* to be differentially expressed in our dataset and validate the activation of this regulatory pathway (Fig 3C and S3E). We have now added a more in-depth description of these results (line 218-227).

Additionally, we investigated the activation of SREBP2 at protein level through detection of its cleaved form after treatment with both PCZ and PIT (Fig 3D, E). We further confirmed predominant nuclear localization of the activated form, consistent with its functional role (Fig 3F, G). SREBP2 is the main transcription factor (TF) responding to cholesterol deficiency in the cell. In line with this, we did not observe any cleavage of SREBP1 after treatment suggesting that SREBP1 is not involved in the response to cholesterol deficiency caused by PCZ or PIT. This is consistent with literature suggesting that SREBP1 is mainly involved in fatty acid synthesis, while SREBP2 is more directly involved in cholesterol metabolism (e.g. Horton et al. J Clin Invest (2022)). These results are described in the text in lines 228-239.

Moreover, we conducted siRNA experiments targeting *HMGCR* to mimic the on-target mechanism-of-action of PIT. siRNA inhibition of *HMGCR* together with PCZ treatment resulted in reduced *HMGCR* protein levels and enhanced cleavage (activation) of SREBP2 (Fig. 4A–C and S3G). These findings further support our hypothesis of a dual impact on cholesterol homeostasis and have been incorporated into the new Figure 4 (lines 241-251).

In addition to incorporating further experiments that support the proposed effect on cholesterol metabolism, we have revised the Results and Discussion sections to present our conclusions more cautiously. To avoid potential confusion with our own findings, we have also removed the previous Figure 3D from the manuscript.

Comment 2

In Figure 5C and 5D, differentiation from MES to ADRN is estimated using transcriptome. However, in Figure 5D, they did not detect the changes expected from the *in vitro* results in Figure 4. Although the authors discuss this straightforwardly in the text (lines 342, 343), there is no detailed discussion of this result. This result should be experimentally examined by single-cell analysis or immunostaining. In immunostaining, markers such as *PHOX2B*, *SOX9*(MES), *LGR5*(MES), *TH*(ADR), and *MYCN* have been reported (Manas et al., Science Advances 2022). The combination of these markers with immune cell markers may enable a detailed description of the molecular mechanisms described above.

Thank you for highlighting this intriguing aspect regarding ADR and MES-like cell states/phenotypes.

In response to the reviewer's comment, we examined NB cell state markers at single cell level *in vivo*. We performed additional staining of tissue sections from the *in vivo* study presented in Fig 6E, F and EV2C using the Phenolmager system (Akoya Biosciences). We have stained for the following NB and cell state markers: NCAM/CD56 (NB), PHOX2B (ADRN), TH (ADRN), SOX9 (MES). Notably, the results indicated an increased fraction of PHOX2B⁺ cells following treatment. SOX9 also showed a trend to increase after treatment, but due to high inter- and intra-tumoral heterogeneity, this was not statistically significant. We further detected co-expression of PHOX2B or TH with SOX9. This observation could indicate a continuum between ADR and MES cell phenotypes at single cell level but additional studies are needed to fully clarify this. The new results are presented in **Figure 6E, F and EV2C (line 316-325)**. The selection of markers is limited and the results thus only represent a limited subset of state-specific identifiers. As the reviewer correctly points out, we did not observe a downregulation of MES-like signatures *in vivo*. This finding is in line with previous studies reporting differential responses of ADRN and MES cell states *in vitro* versus *in vivo* (Thirant *et al.*, *Nat Commun*, 2024). We have now included a more detailed discussion of this discrepancy between *in vitro* and *in vivo* data in the revised manuscript (**lines 402-413**).

Immune cell markers were not included due to the use of NSG mice, which lack T, B, and NK cells. We argue that alternative models are better suited to study immune cell interactions with NB phenotype markers.

We previously (Manas *et al.* 2022) investigated NB cell states in all models used in the present study. We found upregulation of the MES-like state as a response to chemotherapy (Manas *et al.* 2022). This contrasts with the upregulation of the ADRN state seen in the current study, indicating that the shift toward an ADRN-like state is not a general response to treatment in these models. It is important to note that the presence of ADRN or MES-like states in our study is inferred through enrichment analysis, as described in the Methods section (**lines 846-851**). Therefore, these data reflect relative changes in cell state composition and should not be interpreted as absolute quantification.

We believe that the manuscript has been strengthened by the reviewer's comment, and we acknowledge the complexity of NB cell states and their associated markers across different experimental contexts and model systems.

Comment 3

Related to 2), increased clonal diversity is a major molecular mechanism of chemotherapy resistance in COJEC (Manas et al., Science Advances 2022), and whether PCZ+PIT can suppress clonal diversity should be examined.

We agree that the development of treatment resistance and its relationship to clonal diversity and cell states warrants further investigation. However, increased clonal diversity, as measured and reported in Manas *et al.* 2022, was not associated with chemotherapy resistance. Rather, clonal diversity increased as a function of time, independent of treatment pressure. Similar results have been reported by Turati *et al.* (*Nat Cancer*, 2021) and Grossman *et al.* (*Cancer Disc*, 2024). Therefore, this study focuses on transcriptional, proteomic, and lipidomic changes caused by PIT and PCZ.

Comment 4

Although the authors used only MYCN-amplified neuroblastoma in their experimental data, they treated both MYCN-amplified and non-amplified neuroblastoma as high-risk neuroblastoma in

their *in silico* data analysis. They also discuss the possibility that the combination of these drugs may be effective in MYCN non-amplified neuroblastoma (lines 459-461), but there is no experimental data to support this. It remains to be clarified whether the combination of these agents is effective only in MYCN amplified NBs or also in high-risk MYCN non-amplified NBs. At least, they should test whether *in vitro* drug sensitivity differs between MYCN amplified NB and non-amplified NB cell lines. It should also be examined whether MYC expression levels, transcriptional activity, cholesterol deficiency and ADRN-like differentiation are detected in MYCN non-amplified NB- cell lines sensitive to PCZ+PIT. In the data analysis, Fig 4D and Fig S5A should be reanalyzed separately for each MYCN status.

The reviewer is correct that the original *in silico* analyses included both MYCN and non-MYCN tumors as the aim was to identify repurposing candidates for high-risk NB regardless of MYCN status. Risk stratification was based on patient characteristics available in each dataset (Table S1).

Following the reviewer's valuable suggestions, we conducted additional experiments showing that non-MYCN-amplified NB cell models exhibited similar reductions in viability after single and combination treatments as MYCN-amplified NB models (Fig. EV1C, D; lines 173-176). Moreover, Western blot analysis confirmed that MYCN protein levels remained stable post-treatment in MYCN-amplified models (Fig. 3D, E and Fig 4B, Fig S3G), while early responses (48h) included SREBP2 nuclear localization (Fig 3F, G), indicating that disruption of cholesterol homeostasis is the primary effect of PCZ+PIT treatment. These mechanistic insights are detailed in Figures 3 and 4 and discussed in the revised text (lines 426-433).

Based on this evidence, we argue that comparisons in publicly available datasets are appropriately made based on risk status, independent of MYCN amplification. We have therefore retained the original SEQC498 and TARGET161 analyses without subdivision by MYCN status (Figures 5D and S5A, B).

Comment 5

As the authors recognize, the greatest limitation is that the method of administration is only effective in the context of intratumoral treatment; drug repurposing is beneficial because it allows the use of drugs with known side effects and pharmacokinetics, but the results suggest that pharmacokinetics need to be improved. Thus, there is a need to synthesize new compounds with enhanced tumor accumulation using these drugs as seed compounds, and to reevaluate the safety and side effects of these new compounds. In other words, the usefulness of drug repurposing discussed in the introduction and discussion of this manuscript is likely to be compromised. The authors do not discuss methods to increase tumor accumulation without changing the chemical structure of PCZ or PIT, although such methods exist. In addition, these drugs are already in clinical use for the treatment of other diseases, and it has not been discussed whether the maximum dose and route of administration of these drugs in humans is similar to the dose and route of administration in the mouse model. Furthermore, *in vitro*, prior treatment with the PCZ+PIT combination showed enhanced cytotoxic effect (Fig. 4E), while *in vivo*, COJEC and PCZ+PIT were administered simultaneously (Fig. 5E-H). Considering that drug-drug interactions increase the risk of unexpected side effects, the combination of five drugs and PCZ+PIT is likely to increase the risk of side effects other than weight loss. The efficacy of the PCZ+PIT followed by COJEC treatment should be tested *in vivo*.

We thank the reviewer for the suggestion to include a discussion about alternative methods for increasing tumor concentrations of the new drugs. Indeed, in high-risk neuroblastoma this approach is already used by for example disialoganglioside (GD2)-targeted immunotherapy and CAR-T cells (Del

Bufalo *et al.* 2023). We have now added a few sentences about these approaches in the manuscript (lines 443-446).

We acknowledge the difficulty in directly comparing active drug concentrations between humans and mice. In this study, we based our dosing on prior publications and ongoing clinical trials, selecting 5 mg/kg daily for both drugs. For pitavastatin, previous pediatric studies have used 1–4 mg daily, demonstrating safety from age six (e.g., Braamskamp *et al.* 2015; Yazici *et al.* 2025; Harada-Shiba *et al.* 2018). For prochlorperazine, adult doses typically range from 5–10 mg, while pediatric studies (e.g., NCT03984045, NCT06182098) have used 0.15 mg/kg (max 10 mg).

Regarding administration, we initially tested intraperitoneal injection to mimic i.v. delivery but found it ineffective. To confirm treatment effects, we opted for intratumoral injection and have now expanded the discussion on the clinical limitations of this route (lines 434-437).

We agree that applying PCZ+PIT before COJEC *in vivo* would ideally replicate our *in vitro* findings. However, in the clinical setting, chemotherapy begins immediately upon diagnosis of high-risk NB. To avoid an unrealistic treatment delay, we designed our study with a 2-day pre-treatment phase before chemotherapy. We now clarify this design in the revised text (lines 404-408).

Minor comments

Comment 1

Without reading the main text, it is impossible to understand the meaning of "dual-hit mechanism" and "phenotype switch" described in the abstract. These should be changed to more specific descriptions.

We thank the reviewer for the comment. In the abstract, we have changed the “dual-hit” mechanisms to: “The drug combination altered cholesterol metabolism through two different mechanisms ...” (lines 34-35). We have also changed the “phenotype switch” term to “...induced a phenotypic change towards an adrenergic state *in vitro*, which was associated with enhanced chemosensitivity.” (lines 35-36)

Comment 2

2) In Fig. 1 legend, too, "dual-hit mechanism" is not immediately understandable to readers. In particular, the "dual-hit hypothesis" is used to explain completely different etiologies in the fields of Parkinson's disease, schizophrenia, and cardiovascular disease, and is likely to cause confusion. The "dual-hit" hypothesis in these fields defines a first hit and a second hit, which is very different from the way the term "dual-hit" is used in this manuscript. In particular, since LDL-C is associated with the first hit in cardiovascular disease, the reader may misunderstand the meaning of cholesterol control in this manuscript as being related to LDL-C at the beginning of their reading. It is appropriate to simply state, such as cholesterol deficiency.

Again, we thank the reviewer for bringing this to our attention. We have changed this in the manuscript including Figure 1 legend and we now refer to our strategy as a *double-strike* instead of *dual-hit* (line 11 in Figure legends). The term is also described in the Results (lines 270-271).

Comment 3

The details of the drug selection process in lines 141-148 should be provided. How many drugs were initially identified as candidates, how many of each were eliminated, and for what reasons, and finally the 12 compounds in Table 1 were selected.

We thank the reviewer for highlighting the benefit of greater clarity with respect to drug selection. We address the concerns through the addition of text describing the drug selection rationale in more detail (lines 107-113, 132-142).

Comments:

4) Table S3 IC50 concentration units should be described.

Thank you for identifying the missing unit, we have now added μM to table S3.

5) The drug concentration in Figure 2D should be described.

Thank you for the comment. We have now added the drug concentration to Figure 2D and figure legend, as well as in methods (line 601).

6) The meaning of QD in Figure 2F should be described.

Yes, this has been added to line 36 in Figure legends.

7) The calculation method of total efficacy volume in Figure 2B should be described.

Yes, this has been added to lines 22-23 in Figure legends.

8) In Figure 2H, there are unclear characters under "days".

Thank you for noticing, this has now been corrected.

9) In Line 298, "undifferentiated MES-like signatures" may be a new term defined by the authors to equate the transitional subtype of Yuan et al. with the MES subtype of Manas et al. . However, Yuan et al. distinguish transitional subtype from MES subtype (Yuan et al., Cell Rep 2022). The new term should be clearly stated and defined.

We have chosen to use the term "undifferentiated MES-like" to refer to both the strictly mesenchymal subgroup (as described in van Groningen *et al.*, 2017; Patel *et al.*, 2024) and undifferentiated neural crest-like cells observed in other studies (e.g., Boeva *et al.*, 2017; Jansky *et al.*, 2021). Our previous work (Manas *et al.*, 2022) has shown that at least some MES cells exhibit an immature embryonal phenotype. However, we did not intend to include the transitional subtype defined by Yuan in this subgroup as this is, as the reviewer points out, a separate group.

We thank the reviewer for highlighting this text, and we have now rewritten it to clarify (lines 277-283): *GSEA showed enrichment of several ADR gene signatures derived from patient tumors (Bedoya-Reina, Yuan, and Patel) and an integrated signature from multiple datasets (Manas) (Fig. 5A, B). Further, treatment decreased one undifferentiated MES-like signature (Manas) (Fig. 5A, C). The decrease in the recently described transitional subtype of aggressive bridging cells (Yuan) (Fig. 5C), as well as the strong decrease in SYMP-Patel signature (defining actively proliferating NB cells of mainly ADR subtype) 282 (Fig. 5A), likely reflects decreased NB cell proliferation following treatment.*

10) It is not clear to the reader what the SYMP-Patel signature in Line 298 means.

Thank you, we have now clarified this term in the text (line 282): *...(defining actively proliferating NB cells of mainly ADR subtype)...*

11) The possible relationship between the dual-hit mechanism and the phenotype switch should be discussed.

This is an interesting point and we agree that it warrants further investigation. We have mentioned the possible connection between cholesterol deregulation and cell state transition in the Discussion (Ref 45, lines 419-422). However, as this relationship has not been directly tested in the present study, we have chosen not to expand on it extensively. That said, recognizing its relevance to NB, we have now clarified its potential implications (lines 422-425):

The latter relationship between cholesterol homeostasis and cell state changes could (at least partially) explain the observed phenotypic change in NB upon PCZ+PIT targeting, but further studies will be necessary to confirm a causation.

Referee #2 (Remarks for Author):

The manuscript "Repurposing statins and phenothiazines to treat chemoresistant neuroblastoma" by Radke et al explores the potential to use statins and phenothiazines in the treatment of MYCN-amplified neuroblastoma. The authors applied in silico prediction to identify approved drugs for repurposing against neuroblastoma which identified statins and phenothiazines. The work validates these drugs as single treatments and in combination, as well as together with clinically relevant chemotherapy protocol, in MYCN-amplified tumor models in vitro and in vivo. The manuscript is well written, the data is solid and convincing, and clearly presented, and the discussion is relevant and appropriate. The work has strong translational value, investigating drugs already used in the clinic. The findings would be of great interest to the neuroblastoma and cancer community.

I do have some recommendations, comments, and questions for the authors, outlined below.

Comment 1

The study uses neuroblastoma organoid models in vitro, the authors describe that the cells are cultured as free-floating spheres in serum-free medium with the addition of EGF and bFGF, is anything known in regard to the cells' clonal heterogeneity? To differentiate them from cell line models. How do the cells grow in a plate, do they attach or do they form spheroids? How was the viability determined after 7 days if the cells grow in spheroids/clusters? Since CellTiterGlo is described to poorly penetrate 3D spheroids.

We thank the reviewer for this comment. As demonstrated in our previous work (Karlsson et al., *Nat Commun*, 2024; Ref. 30), our NB tumor organoids comprise a heterogeneous population of cells. We now also emphasize this heterogeneity more clearly in the revised text (line 147).

Regarding the growth patterns, NB organoids grow as free-floating spheres and do not attach. Formation of spheroids is achieved via NB cells self-assembly and proliferation. We kindly refer the reviewer to our previous publication (Persson et al. *Sci Rep*, 2017, ref 28) for more information on establishing these organoids and their growth patterns.

Indeed, CellTiterGlo does not penetrate large organoids and tissue pieces well. However, from correspondence with the CellTiterGlo supplier representative (Promega), organoids up to 800µm result in similar penetration when comparing standard CTG and the 3D product. From our in-house tests, we have found that the addition of CellTiterGlo followed by 2min shaking of the plate on an orbital shaker at around 350rpms and gentle mixing at <50rpms at the time of the 10 min incubation results in fully penetrated organoids and proportionate signal as in CellTiterGlo 3D. We now add these protocol details in lines 589-590.

Comment 2

siRNA experiments downregulating HMGCR, in combination with phenothiazines (e.g., PCZ) would strengthen the study in terms of further confirming the mechanism of action for the synergy.

We thank the reviewer for this comment. In response, we have performed the suggested experiment. siRNA-mediated inhibition of *HMGCR* gene expression resulted in downregulation of the enzyme at protein level and increased cleavage (activation) of SREBP2 (the main regulator of cholesterol production) (Fig 4B, C). Additional treatment with PCZ further reduced NB viability and increased apoptosis (cleaved PARP) (Fig 4A, B and Fig S3G). We believe these results further strengthen our hypothesis of a double-strike on cholesterol homeostasis, and we have included the results from this experiment in the new figure 4 (lines 241-251).

Comment 3

As the authors point of themselves, the drugs are only tested in MYCN-amplified models in vitro and in vivo, please make this clearer in the conclusions of the paper, for example in Fig 6.

We have now added analyses of drug efficacy in non-*MYCN* amplified NB cell lines. Treatment affected the viability of these models and drug synergy was detected in SK-N-AS cells (lines 173-176, Fig. EV1C, D). Additionally, we investigated downstream effects of the combination treatment in our *MYCN*-amplified NB organoids and found that *MYCN* protein expression was not significantly affected after treatment (Fig 3D, E, line 235-239), suggesting that *MYCN* is not driving the mechanism-of-action upon combination treatment.

The original predictions were performed in both *MYCN* amplified and non-amplified tumors (lines 107-113, Table S1) to identify potential treatments against aggressive high-risk NB regardless of *MYCN* status. While most of our experimental validation was conducted using *MYCN*-amplified models, we found no evidence suggesting that the drug combination is effective only in *MYCN*-amplified tumors. On the contrary, our results indicate *MYCN*-independent effects.

We have updated the discussion in accordance with the new results (lines 426-433). The old figure 6 has been modified and moved to the synopsis.

Comment 3

Please specify what is shown in each figure, i.e., mean, measurement of variation, number of experiments, etc. Also, please clarify what is shown in Figure 2B to the left, define MSA synergy score and efficacy score in the figure legend (MSA is explained in the text but not efficacy score).

We thank the reviewer for this comment. We have now improved our figure legends.

Comment 4

How was synergy defined? In SynergyFinder it is suggested with >10. The authors claim that the combinations induce synergy in Fig. 2B-C and S2B, which is mostly true for 7 days, but not 3 days according to the limit suggested by SynergyFinder, more correctly would be to state additive or synergetic effects, or specify that the synergetic effects are more pronounced at the longer time point. Furthermore, do the authors have any possible explanation for the improved synergetic effects for longer incubation time?

We agree with the reviewer and we have changed the text to “additive or synergistic ZIP scores” in Results line 162 and Discussion lines 380-381.

The improved synergistic effects over time (mainly visible in LU-NB-2 cells) could be explained by an initial effect of PCZ+PIT combination on cholesterol metabolism that will precede (in time) the effect on viability (as measured by CellTiterGlo). Different cell and tumor models may be differentially sensitive to metabolic disturbance and cholesterol deficiency. However, we have not tested this hypothesis.

Minor comments

Comment 6

The authors propose that the lack of in vivo activity is due to first pass metabolism. Intraperitoneal injection (ip) bypasses first-pass metabolism from the gut, but is still subject to first-pass metabolism by the liver, so it is a plausible explanation, however if so, only owing to metabolism in the liver.

We thank the reviewer for pointing this out. We have clarified this limitation in the discussion (line 434-446).

Comment 7

I believe that the authors might have done a mistake with the formula for calculating in vivo tumor volume, given at row 620 and 650 and written: $V = \pi(\text{length})(\text{width})^2$.

We thank the reviewer for noticing the mistake, the corrected formula is now in line 619.

Referee #3 (Remarks for Author):

Manuscript "Repurposing statins and phenothiazines to treat chemoresistant neuroblastoma" by Katarzyna Radke and colleagues addresses an unmet clinical need in the area of childhood cancer neuroblastoma, i.e. the identification of targetable pathways to eliminate chemo-resistant tumor cells.

The authors use in silico prediction tools to identify statins in combination with phenothiazins. The combination treatment shows synergistic effects in 3D cultures in vitro and in patient-derived mouse models in vivo. The authors further claim that the statin and phenothiazine drug combination leads to a switch towards adrenergic tumor cell state, which is more sensitive to chemotherapeutics.

Overall, the study is well designed and of high relevance, especially for patients with neuroblastoma. However, there are several important controls and experimental replicates missing, which limits interpretability of data and topics that need careful considerations and revision by the authors:

Major:

Comment 1

The strategy applied for differential gene expression analysis using the datasets listed in supplemental table 1 seems arbitrary, comparing a mixture of different parameters. It seems that the authors have for each gene set applied a different approach: they compared on the one hand patients that are alive against dead (all stages!), or MYCN amplified against nonMYCN amplified, low risk etc. It would be much more informative if the analysis would be restricted to one specific question, e.g. within high-risk neuroblastoma MYCN amplified vs non-amplified; or low-risk vs high-risk stages; or within high-risk alive vs dead.

We thank the reviewer for highlighting the benefit of greater clarity on experimental study design with respect to generation of an NB transcriptional query signature used in connectivity mapping (DGEM algorithm). Since the predictions received from the current comparisons within each dataset is the foundation of the entire study, it is not possible to change the comparisons at this point since all subsequent experiments are performed with compounds predicted through this method. We have tried to be as transparent as possible with the different contrasts and patient inclusions of the different datasets in Table S1 and methods (lines 496-503).

However, we want to stress that for each gene set we applied *the same rationale*, that is to define a query gene signature that encapsulates the molecular descriptors of *worsening* disease. The reviewers correctly surmise that within the context of NB biology this could be represented as *MYCN* amplified vs. non-*MYCN* amplified disease, low-risk vs. high-risk disease, and alive vs. dead. The intention is to survey multiple facets of NB biology within connectivity mapping, cast the net wide to discover drugs likely active in NB irrespective of subtype. By this approach, we avoid biasing outcomes with prior knowledge, taking a more target-free approach to discovering novel therapies. We have addressed the reviewers' concerns by clarifying the rationale more explicitly in the Results (lines 107-113).

Comments:

2. Why did the authors only validate their findings in MYCN amplified NB models? It will be very important to add MYCN amplified models for in vitro and in vivo validation.

We agree with the reviewer that our work would benefit by also including non-*MYCN* amplified models. We have thus performed viability assays of single drugs and the combination in two non-*MYCN* amplified cell lines: SK-N-AS and SK-N-SH. Both PCZ and PIT decreased viability in the non-*MYCN* amplified NB cells and the combination was synergistic in SK-N-AS cells (Figure EV1C, D, lines 173-176).

Furthermore, our new analysis of downstream protein expression in *MYCN*-amplified NB organoids revealed that *MYCN* was not directly involved in the response to PCZ or PIT treatment (Figure 3D, E; lines 235-239). Based on these findings, we proceeded with further experiments using our *MYCN*-amplified NB models.

We hope that the inclusion of data from non-*MYCN*-amplified cell lines, along with our downstream characterization of effector proteins, provides a clearer rationale for our choice of NB models in the manuscript.

3. It is important to recognize that 60% of high-risk NB cases do not carry a MYCN amplification and will therefore likely not benefit from the proposed treatment.

We thank the reviewer for this important comment. We have found no indications that non-*MYCN* amplified tumors would be unresponsive to the proposed treatment. As described above (comment 2), both PIT and PCZ showed individual and combined effects on cell viability in *MYCN* and non-*MYCN* amplified tumor cells (Figure EV1C, D, lines 173-176). Furthermore, the initial predictions as well as other bioinformatic analyses of patient material were performed using patient data that included both *MYCN* subgroups.

Mechanistically, our results indicate that the primary effect of treatment is cellular cholesterol deficiency. *MYCN* protein levels were not significantly altered following treatment (Figure 3D, E; lines 235-239), suggesting that the response is not dependent on *MYCN* status.

Therefore, we believe that the proposed combination therapy could be a viable option for both *MYCN*-amplified and non-amplified tumors. We have expanded the discussion of this point in the revised manuscript (lines 426-433).

4. It should be clarified that a ADRN and low risk signature are very different and this should be considered in the analysis and interpretation accordingly.

We agree with the reviewer that ADR gene expression signature and the gene expression associated with low-risk NB patient group are different and that this should be clear in the manuscript. In our analysis we have used the ADR gene signatures that have been described as cell state specific in previous publications (outlined in Table S4). The rationale for also investigating the association between the transcriptional treatment response (genes upregulated after treatment) and gene expression in NB patients of the low(er)-risk groups was that the original predictions were based on transcriptional datasets from NB tumors of different risk-status (Fig 5D and S5A, B, C).

We believe that both analyses are relevant to report. To clarify the differences, we have restructured the manuscript and now present the findings in two separate paragraphs (lines 273 and 284, respectively).

We have also clarified this in the summary sentence of lines 300-302: *Together, our data suggests that PCZ-PIT treatment leads to a phenotypic change towards an ADR cell state in vitro and that the treatment-induced response correlates with transcriptional profiles of low-risk NB.*

5. Sensitivity of Neuroblastoma to Statins is not new and has been reported previously (e.g. PMID: 23579272, PMID: 22759742, PMID: 19739078,...)

It is correct that statins have previously been shown to have an effect in NB. What is novel in this study is the combination of drugs that target cholesterol dependency of cancer cells. We had already included the following three references (Almstedt *et al.*, Girgert *et al.*, Arnold *et al.*; 39-41) for previous studies in NB. We acknowledge that additional relevant studies could be cited. We appreciate the suggestion to include Marcuzzi *et al.*, 2012 (Ref. 42; line 377), as it provides valuable context by describing statin-induced apoptosis via the mitochondrial pathway.

In the end of the discussion, we have removed the word “new” from summary (line 469): “*We identified two classes of drugs which, when administered together, hold promise for patients with chemoresistant NB*”. After a thorough search through the manuscript, we believe that in all other places in the text we are referring “new” or “novel” for the combination (not the individual drugs or classes of drugs).

6. The effect in vivo is mild (delay in tumor growth, no cure), despite intra-tumoral administration. This limits the relevance for translation into clinical trials.

We thank the reviewer for the thoughtful comment regarding translational relevance.

Our choice of PDX model was guided by the clinical need for improved treatment strategies for chemoresistant NB. Therefore, we selected PDX #1, which is highly chemoresistant, as previously described (Manas *et al.*, 2022). Nevertheless, in the initial intratumoral (i.t.) in vivo study, the PCZ+PIT combination resulted in a mild but statistically significant reduction in tumor growth and improvement in survival compared to non-treated controls.

From a clinical perspective, the potential to introduce a novel treatment alongside existing chemotherapy would be highly valuable. Given that induction chemotherapy is standard of care for

high-risk NB patients, we designed the final *in vivo* study to reflect a clinically relevant treatment scenario.

We would like to highlight the data in Figure 6G-J, which demonstrate a markedly enhanced effect on tumor size and suggest the possibility of long-term remission in the group receiving both the PCZ+PIT combination and induction chemotherapy.

That said, we fully acknowledge the limitations and challenges associated with the clinical translation of the compounds investigated in this study. These considerations are now discussed in greater detail in the revised manuscript (lines 434-466).

7. Only one mouse model has been used. It is the understanding of the reviewer, that this is not sufficient proof and it will be necessary to repeat the experiments in a second mouse model.

We thank the reviewer for the relevant comment regarding the use of *in vivo* models.

We provide *in vitro* data derived from multiple organoid and cell line models now representing both *MYCN*-amplified and non-amplified NB.

As previously demonstrated (Manas et al., 2022), the PDX#1 model is highly aggressive, and exhibits pronounced resistance to chemotherapy. Given its refractory nature, we selected this model for our *in vivo* experiments to reflect a clinically challenging treatment scenario.

We agree that including additional *in vivo* models could further support the generalizability of the *in vitro* findings. However, we do not believe that using additional models would substantially alter the main conclusions of the manuscript. In line with the 3R principle and our efforts to minimize animal use, we focused on conducting an intensive and long-term *in vivo* study using a single, highly representative PDX model (Fig. 6G-J).

8. It is not clear whether mouse xenografts were derived from patient-derived 3Dcultures. It this is indeed the case, it is suggested to refrain from indicating these as PDX (patient-derived xenograft), as this term is usually only used in case patient material is directly transplanted into mice.

Thank you for the comment. In our lab, we have established and expanded orthotopic patient-derived NB tumors in NSG mice. From these tumors, we subsequently generated 3D tumor organoids. These NB organoids have been employed in both *in vitro* assays and re-implanted into mice for *in vivo* experiments. We have extensively characterized these models in terms of molecular features and genetic stability as described in our previous publications (Braekeveldt *et al.*, 2015; 2018).

We fully agree that precise terminology is important in this context. However, with all due respect to the reviewer's perspective, we would like to point out that the term "PDX" is widely used in the literature to describe this type of model system, including in scenarios consistent with our study design (e.g., Stewart *et al.*, *Nature*, 2017; Bleijs *et al.*, *EMBO J.*, 2017; Tucker *et al.*, *J. Personalized Medicine*, 2021).

9. Experiments shown in Figure 3 are interesting, but descriptive. No mechanistic investigations were attempted, e.g. by genetic loss and gain of function experiments.

We thank the reviewer for the insightful comment. In addition to gene expression analysis, we performed functional rescue experiments demonstrating that supplementation with mevalonate was able to rescue the growth inhibition induced by PIT, both as a single agent and in combination treatment (previous fig. 3E, now fig. 4D). In contrast, mevalonate supplementation did not rescue

PCZ-treated organoids, suggesting a distinct mechanism of action for PCZ compared to PIT. For PCZ we showed co-localization of cholesterol and lysosomes suggesting accumulation of cholesterol, also shown by filipin-staining (previous Fig 3F-G, now fig. 4E, F).

In response to the reviewer's comment, we have conducted additional mechanistic investigations that further support and strengthen our findings. SREBP2 is the main transcription factor (TF) responding to cholesterol deficiency in the cell and we examined the activation of SREBP2 by detecting the cleaved, active form of the protein after treatment with both PCZ and PIT (Fig 3D, E). We also confirmed prominent nuclear localization of the activated form (Fig 3F, G).

We further performed siRNA experiments by inhibition of the expression of *HMGCR* to mimic the mechanism-of-action of PIT. siRNA inhibition of *HMGCR* lead to downregulation of the enzyme at protein level (indicating on-target effect) and to increased cleavage (activation) of SREBP2 (Fig 4B, C). Importantly, additional treatment with PCZ further decreased NB cell viability in a concentration dependent manner (Fig 4A).

We believe these new results provide strong mechanistic support and further strengthen our hypothesis of a double-strike on cholesterol homeostasis.

10. The authors claim that "treatment of NB organoids with PIT and/or PCZ decreased proliferation and caused compensatory upregulation of cholesterol production through high transcriptional activity of transcription factors SREBF1/2". This has not been proven experimentally.

We thank the reviewer for the opportunity to clarify this point.

We showed an upregulation of the transcriptional activity of *SREBF1* and 2 upon treatment with PCZ and PIT through inference analysis with CollecTri (now Fig 3B and S3F). Furthermore, our transcriptomic data showed that known downstream effectors or activators of *SREBF1* and 2 (e.g. *ACSL1*, *ASCSS2*, *INSIG1*, *FDPS*) were upregulated after PCZ+PIT combination treatment (now Fig3C and S3E). To improve clarity, we have added a more detailed description of these findings to the Results section (lines 218-227) and moved some of the graphs to the main figure.

In additional response to the reviewer's comment, we conducted further mechanistic experiments. Thus, we have now investigated the activated form of SREBP2 through detection of the cleaved form after treatment with both PCZ and PIT (Fig 3D, E, lines 228-234). Additional related findings from our siRNA experiments are discussed in our response to comment 9 (Fig 4B, C, lines 247-249).

Nuclear localization of SREBP2 was also observed after combination treatment (Fig 3F, G, lines 234-235), confirming its activation. In contrast, we did not detect the cleaved form of SREBP1 under the same treatment conditions, suggesting that SREBP1 is not involved in the response to cholesterol depletion caused by PCZ or PIT (Fig 3D). This is consistent with literature which suggests that SREBP1 is mainly involved in fatty acid synthesis whereas SREBP2 is involved in cholesterol pathways (e.g. Horton *et al.* J Clin Invest, 2022).

11. Figure 3D is more suitable for a graphical abstract and distracting from the results presented in this figure.

We agree that it might be confusing to present an illustration of the suggested mechanism in the middle of figure 3. We have now removed this figure.

12. In Figure 4a it will be important to include the combination alone as a comparator. Otherwise, it is difficult to judge what the contribution of the individual treatments are.

Thank you. Figure 4a (now Fig. 5A) includes both the combination (PCZ+PIT) as well as PCZ and PIT as single drugs. All treatment conditions are compared to the non-treated control. We would be happy to further clarify this in the figure legend or text, should the reviewer find it helpful.

13. In the *in vivo* experiments ADR and MES states were only inferred by gene expression data. There is no direct evidence *in vivo* shown. This is a limitation, especially as it is not shown by the authors that MES-like tumor cells exist in their model and are reduced or switch upon treatment. This is important as it is claimed that the combination treatment induces a switch towards ADR-like tumor cells.

Thank you for highlighting this aspect regarding ADR and MES-like cell states/phenotypes. The full existence and potential switch between ADR and MES cell states is still being debated (e.g. Durbin and Versteeg, *EJC Paediatric Oncology*, 2024) and it is not within the scope of this manuscript to provide a full characterization of the cell states. Nevertheless, here we provide an explanation of our approach and novel experiments conducted in response to the reviewer's comment.

We defined cell states by including multiple gene signatures for each of the two originally defined states (list in Table S4). We analyzed gene expression at bulk level, thus it was not possible to detect single MES-like tumor cells, but rather a change in phenotype overall. *In vitro*, we demonstrated a strong upregulation of ADR gene expression after treatment, but we were not able to state if this was caused by an induced phenotypic switch or a by selective killing or inhibition of MES-like gene cells/expression. We agree that this is an important aspect, as also noted by the reviewer in comment 14. To improve clarity in the manuscript, we have now replaced the word "switch" with "change" in relevant instances (e.g. lines 35, 300, 402).

In response to the reviewer's comment, we examined NB cell state markers at single cell level *in vivo*. We performed staining of tissue sections from the *in vivo* study presented in Fig 6E, F and EV2C using the Phenolmager system (Akoya Biosciences). We have stained for the following NB and cell state markers: NCAM/CD56 (NB), PHOX2B (ADRN), TH (ADRN), SOX9 (MES). Notably, the results indicated an increased fraction of PHOX2B⁺ cells following treatment. SOX9 also showed a trend to increase after treatment, but due to high inter- and intra-tumoral heterogeneity, this was not statistically significant. We further detected co-expression of PHOX2B or TH with SOX9. This observation could indicate a continuum between ADR and MES cell phenotypes at single cell level but additional studies are needed to fully clarify this. The new results are presented in Figure 6E, F and EV2C (line 316-325). The selection of markers is however limited and the results thus only represent a limited subset of state-specific identifiers.

As the reviewer correctly points out, we did not observe a downregulation of MES-like signatures *in vivo*. This finding is in line with previous studies reporting differential responses of ADRN and MES cell states *in vitro* versus *in vivo* (Thirant *et al.*, *Nat Commun*, 2024). We have now included a more detailed discussion of this discrepancy between *in vitro* and *in vivo* data in the revised manuscript (lines 402-413).

We previously (Manas *et al.* 2022) investigated NB cell states in all models used in the present study. We found upregulation of the MES-like state as a response to chemotherapy (Manas *et al.* 2022). This contrasts with the upregulation of the ADRN state seen in the current study, indicating that the shift toward an ADRN-like state is not a general response to treatment in these models.

It is important to note that the presence of ADRN or MES-like states in our study is inferred through enrichment analysis, as described in the Methods section (lines 846-857). Therefore, these data

reflect relative changes in cell state composition and should not be interpreted as absolute quantification.

We believe that the manuscript has been strengthened by the reviewer's comment, and we acknowledge the complexity of NB cell states and their associated markers across different experimental contexts and model systems.

14. How can the authors distinguish a switch from selective killing of MES-like cells? Lineage tracing experiments would be required to demonstrate this.

We agree with the reviewer that it is not possible to determine if the cells are changing phenotype or if there is selective killing of subpopulations based on our data. Lineage tracing experiments with single cell sequencing would be needed to examine this. We have expanded the discussion about this limitation (lines 408-413).

Minor

1. The discussion should be revised considering the above comments to avoid any overstatements, e.g. "As expected, gene expression and mechanistic analyses..." rigorous mechanistic analysis is missing.

We thank the reviewer for this comment. In response, we have added additional mechanistic studies including siRNA inhibition of *HMGCR*, analysis of cleaved SREBP2, and subcellular localization of SREBP2 after treatment. The new results have been added to Fig 3 and the new Fig 4. We have also revised the Discussion section to avoid overstatements and to provide a more comprehensive discussion on the limitations of the study.

24th Sep 2025

Dear Dr. Bexell,

Thank you for submitting your revised study. We have now received the reports from referees #1 and #2, who also reviewed your responses to referee #3. As you will see below, they are satisfied with the revisions, and I will therefore be able to accept your manuscript once the following editorial concerns are addressed:

1/ Manuscript text:

- Please remove the yellow highlights and only keep in track changes mode any new modification in the text.
- Authors: there is a discrepancy between Erick Muciño (submission system) and Erick A. Muciño-Olmos (manuscript text), please check and adjust. Please indicate on the title page that Neil Thompson is deceased.
- Please delete the "Teaser" after the Abstract.
- We can accommodate a maximum of 5 keywords, please adjust accordingly.
- Please correct the order of the manuscript sections to: Abstract / Keywords / Introduction / Results / Discussion / Methods / Data Availability / Acknowledgements / Disclosure and Competing Interests Statement / References / Main Figure Legends / Tables / Expanded View Figure Legends.
- "Materials and Methods" should be renamed "Methods".
 - o For all mice experiments, please indicate the origin, gender, housing and husbandry conditions, as well as the authority granting ethics approval, provide reference number for approval. Include a statement of compliance with ethical regulations.
 - o Please provide dilutions/concentrations for all antibodies used in the study.
- Please download and fill our Reagents and Tools Table template (.docx), which you can find in our author guidelines: <https://www.embopress.org/page/journal/14693178/authorguide#structuredmethods>. When submitting your revised manuscript, please do not include the Reagents and Tools Table in the Methods section of the manuscript but upload it as a separate file choosing the file type "Reagent Table".
- Data Availability section: please include here the links to the unprocessed deposited datasets generated in this study, including lipidomics data.
- Please merge Funding with Acknowledgements. Remove Biorender from Acknowledgements, and add a dedicated "Graphics" section to the Methods following this format:
Graphics:
(some of the... OR Figure #... OR synopsis) Graphics were created with BioRender.com.
- Please remove the Authors Contributions from the manuscript and use the free text boxes beneath each contributing author's name in our system to add specific details on the author's contribution.
- Please rename "Competing interests" to "Disclosure and competing interests statement".
- References: please correct the formatting to alphabetical order, up to 10 author names listed followed by et al.

2/ Figures:

- Figures S1-S5 should be made EV figures. Alternatively, they could be added to the appendix, together with the legends, which should be added to the appendix file, underneath the figures.
- The file with the supplementary tables should be renamed "Appendix" and uploaded as a PDF. A table of contents should be added, including page numbers. Please correct the nomenclature to "Appendix Table S1" etc. and, if applicable, to "Appendix Figure S1" etc.
- There is a callout for Supplementary Data 1 in the text, please correct.
- Please address the queries from our data editors in the figure legends:
 1. Please indicate the statistical test used for data analysis in the legends of figures 3E, G; 6I, S3D, S4
 2. Please note that the box plots need to be defined in terms of minima, maxima, centre, bounds of box and whiskers, and percentile in the legends of figures 2G, 4GM 5B, C; 6I, EV1 E, EV2 A, D; S2 B, S4
 3. Please note that information related to n is missing in the legends of figures 5B, C; EV1 E, EV2 A, D; EV2 B, S2 B, S3 B, D, G; S4
 4. Your manuscript contains error bars based on n=2 in figure 2E. Please use scatter blots showing the individual datapoints in these cases. The use of statistical tests needs to be justified.
 5. Please note that the error bars are not defined in the legends of figures 4A, C, F; 5E, EV1 C, EV2 B, S1 B-D; S3 B, D, G
 6. Please note that scale bar and its definition are missing for figure EV2 C.
 7. Please note that the yellow arrows are not defined in the legend of figure 4E. This needs to be rectified.

3/ Thank you for providing Source Data. Please check whether there is a mislabeling between Fig. 3F and 3G.

4/ Checklist: please indicate the manuscript number in the top left corner.

5/ Thank you for providing the Paper Explained, please include it in the manuscript text file.

6/ I introduced a minor edit in your synopsis, please let me know if you agree or amend as you see fit:

"Over half of children with high-risk neuroblastoma (NB) exhibit treatment resistance to current therapies. Combining two clinically approved drugs, phenothiazines (PCZ) and pitavastatin (PIT), with standard-of-care chemotherapy was shown to enhance treatment efficacy in chemoresistant NB.

- In silico predictions, including machine-learning analyses, of NB gene expression profiles identified PCZ and PIT for drug repurposing.
- The PCZ+PIT combination led to drug synergy, NB cell death, and cholesterol deficiency in NB organoids.
- The combination of PCZ+PIT induced NB cell state changes in vitro leading to enhanced sensitivity to chemotherapy.
- In vivo treatment with PIT+PCZ in addition to standard-of-care chemotherapy led to decreased tumor size and extended survival in a chemoresistant patient-derived xenograft model of high-risk NB."

Thank you for providing a nice visual abstract. Please resize it to a PNG file 550 px wide x 300-600 px high and make sure the text remains legible. A cropped portion of this image will serve as thumbnail for the table of content on our webpage.

7/ As part of the EMBO Publications transparent editorial process initiative (see our Editorial at <http://embomolmed.embopress.org/content/2/9/329>), EMBO Molecular Medicine will publish online a Review Process File (RPF) to accompany accepted manuscripts.

This file will be published in conjunction with your paper and will include the anonymous referee reports, your point-by-point response and all pertinent correspondence relating to the manuscript. Let us know whether you agree with the publication of the RPF.

I look forward to receiving your revised manuscript.

Yours sincerely,

Lise Roth

***** Reviewer's comments *****

Referee #1 (Remarks for Author):

suitable for publication

Referee #2 (Remarks for Author):

I am pleased with the responses and revisions and believe that the manuscript has improved significantly with the new data and changes. I consider it now suitable for publication.

The authors addressed the remaining editorial issues.

14th Nov 2025

Dear Dr. Bexell,

Thank you for submitting your revised files. I am pleased to inform you that your manuscript is accepted for publication and is now being sent to our publisher to be included in the next available issue of EMBO Molecular Medicine!

With kind regards,

Lise Roth
